# Quantifying the role of variability in future intensification of heat extremes

Claudia Simolo [1] & Susanna Corti [1]

Heat extremes have grown disproportionately since the advent of industrialization and are expected to intensify further under unabated greenhouse warming, spreading unevenly across the globe. However, amplification mechanisms are highly uncertain because of the complex interplay between regional physical responses to human forcing and the statistical properties of atmospheric temperatures. Here, focusing on the latter, we explain how and to what extent the leading moments of thermal distributions sway the future trajectories of heat extremes. Crucially, we show that daily temperature variability is the key to understanding global patterns of change in the frequency and severity of the extremes and their exacerbation in many places. Variability accounts for at least half of the highly differential regional sensitivities and may well outweigh the background warming. These findings provide fundamental insights for assessing the reliability of climate models and improving their future projections.

Global warming has been advancing rapidly since the beginning of the industrial era, exposing humans and the environment to ever-increasing risks[1,2]. The magnitude and pace of climate change, as well as their potential impacts, vary widely across regions[1,3,4] because of both the multiple physical processes that are triggered by increased greenhouse forcing and the inherent regional vulnerabilities. Critical areas exist where average temperatures have already surpassed the most conservative global limits laid out in the 2015 Paris Agreement[5] and the extremes have changed dramatically[2,6–9], the Arctic region being one prominent example. Even worse, climate models predict that these changes will accelerate further with rising levels of warming, stressing the urgency of drastic mitigation actions[10–12]. Despite the current uncertainties in scenario projections[1,12], extreme events distinctly show robust and very heterogeneous changes around the globe. In some places heat extremes are predicted to hugely increase in number and become routine in the next few decades. In others they will rise in severity much faster than what projections of mean climate conditions would suggest, with harmful consequences for humans and ecosystems.

The question of whether and how these changes originate from the background regional and seasonal warming is a very vexed one[6,13–22] and involves both statistical and physical aspects. Amplification mechanisms are likely rooted in the complex Earth-system interactions that unfold against the backdrop of global warming. Among these, large-scale atmospheric changes (e.g., in thermal gradients and circulation patterns) and internal climate feedbacks between soil, snow cover, sea ice and the atmosphere[23–33]. In turn, these processes may induce significant changes in the statistical properties of thermal distributions, namely, their leading moments[34–42].

As growing evidence suggests[14,20,31,34,43], changes in the higher order moments can either amplify or damp the response of the extremes to greenhouse forcing, but the net effect is currently unknown. In the next decades, a widespread decrease in midlatitude temperature variability is expected[31,34,35], which will likely contribute to the rapid weakening of cold-season cold extremes. Furthermore, local-to-regional increases in variability are predicted to enhance the severity of hot events, particularly over tropical lands[21,33,39]. Unfortunately, global data coverage in the recent past is limited, and observational evidence of historical changes in the higher moments is scarce, leaving the matter controversial[15–18].

In addition to higher order changes in thermal distributions, their own native structure has proven relevant in many places[44–48]. Beside variability, intrinsic non-Gaussian properties (e.g., skewness) of atmospheric temperatures can either speed up or inhibit the increase in the chance of unusual hot events with global warming, depending

[1]Institute of Atmospheric Sciences and Climate, National Research Council of Italy, I-40129 Bologna, Italy. e-mail: c.simolo@isac.cnr.it; s.corti@isac.cnr.it

on whether local temperatures exhibit tails shorter or longer than normal at the right end of the distribution[47,48].

Clearly, a comprehensive understanding of the mechanisms behind the differential regional behavior of the extremes would be highly desirable and is not yet achieved. Here we take an important step forward and provide a global-scale quantitative analysis of the impact of the leading distribution moments on the trajectories of heat extremes. Based on the latest scenario projections, we show that the past and future variability of daily temperatures are essential to explain the global patterns of changes in the frequency and severity of the extremes. Namely, changes in variability greatly exacerbate regional differences in their warming rates and cause cold extremes to warm much faster than hot extremes over large areas. On the other hand, intrinsic variability (far more than its future changes) tightly controls the increasing frequency of unusual hot events throughout the globe and most often overrides the background regional warming.

Our findings help unravel the complex heterogeneity issue in the regional response of the extremes to human forcing, laying the ground for diagnosing the root causes. By elucidating the connection between extreme climate trajectories and the statistical properties of thermal distributions, we also provide key elements to better constrain climate model predictions. This is crucial to anticipate the most serious threats of climate change over the coming decades and to support mitigation and adaptation efforts tailored to regional needs.

## Results

Figure 1a shows future changes in global-mean surface air temperature (GSAT) relative to the early industrial era (1851–1900), as these result from scenario simulations endorsed by the Sixth Coupled Model Intercomparison Project (CMIP6)[49], under a range of Shared Socio-economic Pathways (SSPs) and forcing levels (see the "Methods"

section and Supplementary Table 1). The latter comprise a high-mitigation (SSP1-2.6), a middle-of-the-road (SSP2-4.5) and a business-as-usual like scenario (SSP5-8.5). Rates of change in GSAT strictly depend on the given SSP, despite large spreads across models. In the worst case scenario, the Paris (upper) limit of +2K warming above early industrial conditions could be breached before the middle of the century (insets of Fig. 1a), and a +3K warming by 2100 cannot be ruled out even in lower forcing scenarios[50,51].

Here, focusing primarily on these levels of global warming (GWLs), we first investigate changes in the magnitude of the largest hot and cold temperature anomalies of the year or season (the hottest day *TXx* and the coldest night *TNn* respectively), which represent (moderately) extreme events to be expected at the same frequency during the past and the future. Next we deal with the converse, namely, the change in the exceedance probability of large temperature anomalies, whose magnitude is fixed and above a high quantile (the 99th and 99.9th) of the early industrial distributions. Details on the observables and their processing are given in the "Methods" section.

### Differential warming of heat extremes

Enhanced greenhouse forcing may cause a rapid warming of the extremes in many places. Figure 1b, c shows global and land-averaged changes in annual *TXx* and *TNn* at growing levels of warming (regardless of their timings), under alternative SSPs. In line with previous findings[52,53], the extremes exhibit near linear increase with GSAT and quite the same rates of change irrespective of the emission trajectory. Changes over land in both *TXx* and *TNn* are faster than over the global area and robustly exceed the change in GSAT (by about 30 and 70%, respectively, Supplementary Table 2). Continent interiors indeed warm faster than the ocean, consistently with the expected strengthening of land-sea contrasts under transient scenario simulations[36,54].

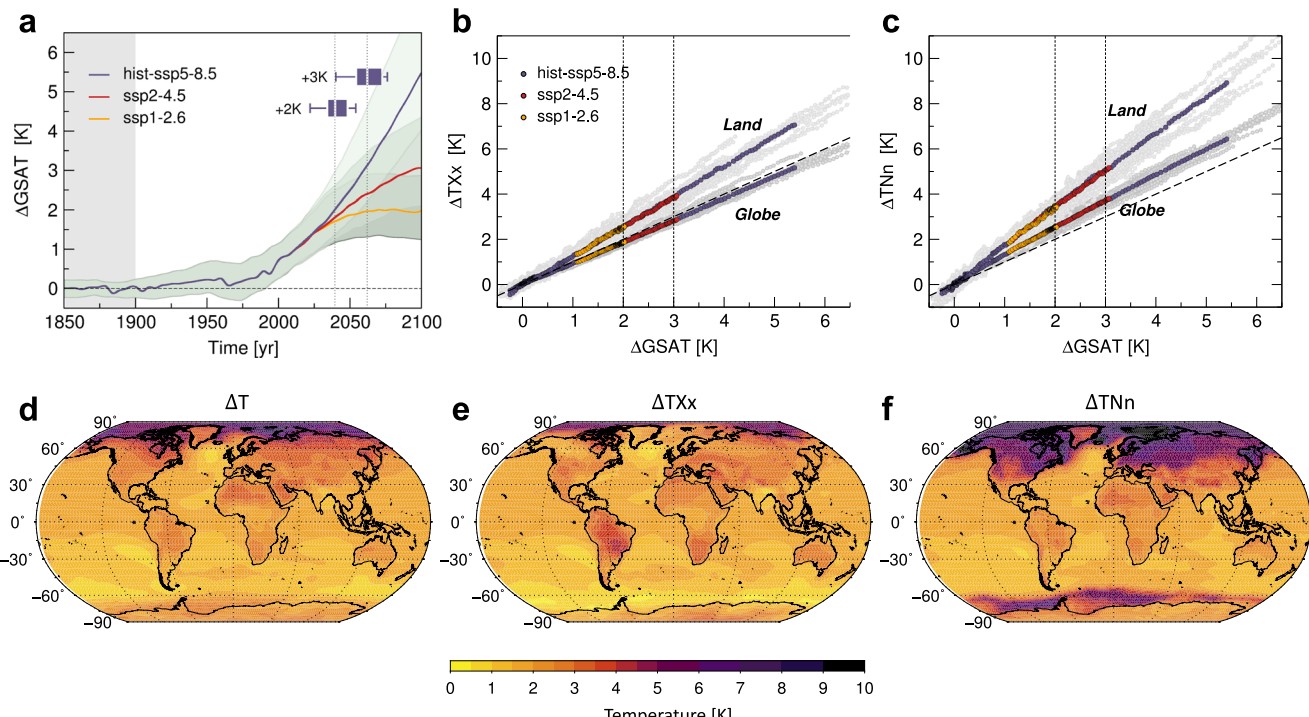

**Fig. 1 | Global changes in mean temperatures and the extremes. a** Changes in global-mean surface air temperature (GSAT) relative to early industrial times (gray shaded background), on the basis of 20 model simulations (Supplementary Table 1) forced by alternative Shared Socioeconomic Pathways, namely SSP1-2.6, SSP2-4.5 and SSP5-8.5. Shaded bands denote the total spread in model projections, solid lines their multimodel mean. Box plots summarize the projected timings for GSAT to reach +2K and +3K above the early industrial mean under SSP5-8.5. They account for the median (inside bar), the upper and lower quartiles (box edges) and the total range (whiskers) of model results. **b, c** Multimodel global and land changes in the hottest days (*TXx*) and the coldest nights of the year (*TNn*), respectively, against changes in GSAT, for the alternative SSPs. Gray dots denote the intermodel spread and dashed lines the identity relation. **d–f** Multimodel patterns of change in annual mean temperatures *T* and the extremes *TXx* and *TNn* respectively, at +2K warming (SSP5-8.5).

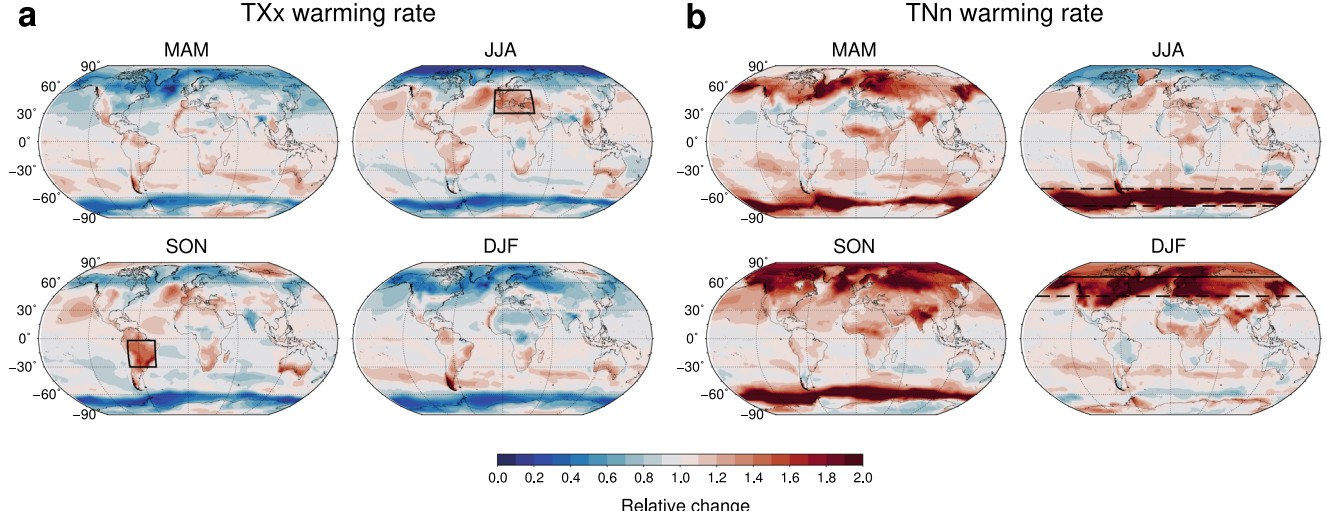

**Fig. 2 | Seasonal extremes.** Multimodel warming rates of **a** the hottest days (*TXx*) and **b** the coldest nights of every season (*TNn*), at +2K warming (in the high-end scenario SSP5-8.5). Namely, $\Delta TXx/\Delta T$ and $\Delta TNn/\Delta T$, where $\Delta T$ is the local changes in annual mean temperatures (Fig. 1d). *TXx* and *TNn* are used here and below to denote seasonal extremes, unless otherwise specified. MAM denotes March-April-May, JJA June-July-August, SON September-October-November and DJF December-January-February. Regions showing a major warming are highlighted.

At fixed levels of warming (Fig. 1d–f and Supplementary Fig. 1), changes in the extremes occur highly unevenly across the globe and depart considerably from changes in mean conditions. For instance, *TXx* anomalies rise faster than the annual mean over several low-latitude and extratropical land regions, whereas *TNn* show the largest increases at middle and high latitudes, with giant amplification over the Arctic.

Warming rates of *TXx* and *TNn* are displayed by season in Fig. 2a, b, where local changes at the +2K GWL are scaled by the respective changes in the annual mean (the local background warming). They evidence regions exposed to a higher risk in a warmer climate, due to faster changes in the extremes than in mean conditions (red-shaded areas). This is the case of the Euro-Mediterranean zone and central areas of South America, where *TXx* warming in the warm seasons may well exceed the local background (by up to 50%, Fig. 2a). At higher latitudes *TXx* changes slow down in every season and reach a minimum in a rather symmetric way over polar regions, where they may be less than half of the changes in the annual mean (and even vanishing in the high-Arctic summer). Conversely, *TNn* warming is much faster than the local background at middle and high latitudes, in all seasons except summer (Fig. 2b). It exhibits the highest rates during winter, with local changes twice as large as the annual mean. In the northern hemisphere, in particular, the strong *TNn* warming extends all over the Arctic and down to midlatitudes. As seen in Supplementary Fig. 2, the rates of change in *TXx* and *TNn* are rather insensitive to an additional increase in GSAT, indicating that the magnitude of the extremes grows near linearly with the local warming.

The highly differential regional and seasonal behavior of the extremes is closely related to the changes in the underlying anomaly distributions. Regional amplifications of *TXx* and *TNn* originate either from a rapid warming of the seasonal mean or from a change in the higher order moments, including variability, or a combination thereof. More precisely, for a given grid point and background warming $\Delta T$, the *TXx* change can be expressed as a function of changes in the first and the higher moments of the *TX* distribution (see Supplementary Note and Supplementary Eq. (1)). Namely,

$$\frac{\Delta TXx}{\Delta T} = \frac{\Delta TXm}{\Delta T} + \left[\frac{(TXx - TXm)}{TXsd}\right]_{EI} \frac{\Delta TXsd}{\Delta T} + \text{h.o.} \quad (1)$$

where $\Delta TXm$ and $\Delta TXsd$ are the changes in the mean and standard deviation respectively (see "Methods" for details), brackets denote the early industrial averages and the h.o. term includes corrections from higher than second order changes as a group. A pure upward shift of the distribution leads the mean and the extremes to increase at the same rate, thereby preserving their mutual distances. By contrast, changes in the higher order moments induce distortions in the native distribution, that can significantly modify changes in the extremes (Supplementary Fig. 3a). Equation (1) and the analogous for *TNn* provide a theoretical scheme for interpreting the regional differences in warming rates and quantifying the roles of the leading moments.

As is shown in Supplementary Fig. 4, the seasonal averages *TXm* and *TNm* increase everywhere in a +2K climate, though very heterogeneously, giving relevant contributions to the changes in *TXx* and *TNn* respectively. For example, a rapid *TXm* increase compared to the local background is seen in the warm season over several land areas coinciding with *TXx* hotspots such as the Euro-Mediterranean zone and South America, among others. In these regions, the excess warming of *TX* anomalies is likely related to long-term trends in soil drying, as this can modify the surface energy balance, with a loss of evaporative cooling and the increase of sensible heat fluxes and thus of temperatures[26,55,56]. Soil moisture-temperature coupling is the strongest in transitional -wet to dry- climates, like that of the Euro-Mediterranean zone showing among the most prominent warming. Furthermore, *TNm* changes, similar to *TNn*, evidence the staggering rate of warming of the Arctic during winter months, which is mainly caused by sea-ice decline and the increase of seasonal heat storage and release by the opened waters[30,34].

Both *TXx* and *TNn,* however, distinctly deviate from the seasonal averages in many places, most notably at mid-to-high latitudes, revealing changes in the higher moments. In polar regions, for example, winter *TXx* show tendencies opposite to those of *TXm* which, unlike the previous ones, warm faster than the annual mean. Clearly, seasonal warming rates cannot account alone for the highly heterogeneous behavior of the extremes. In fact, the fraction of the total spatial variation of *TXx* and *TNn* changes which is explained by the changes in the seasonal averages hardly exceeds 50%, as indicated by the coefficients of determination $R^2_{\text{red}}$ in Table 1. These are obtained by Eqs. (1) and (4) as detailed in "Methods" and are further discussed below.

**Table 1 | Coefficients of determination**

| | $R^2_{red}$ | | $R^2_{full}$ | |
| --- | --- | --- | --- | --- |
| | *TXx* | *TNn* | *TXx* | *TNn* |
| MAM | 0.52 | ≲0 | 0.85 | 0.87 |
| JJA | ≲0 | 0.58 | 0.87 | 0.95 |
| SON | 0.46 | 0.27 | 0.84 | 0.92 |
| DJF | ≲0 | 0.39 | 0.82 | 0.91 |

Fraction of spatial variation of the warming rates of the hottest days (*TXx*) and the coldest nights (*TNn*) at +2K warming, explained by the rates of change in distribution moments through Eq. (1) of the main text. $R^2_{red}$ is the fraction explained by the first moment alone and $R^2_{full}$ is the fraction explained by the first two moments (excluding skewness and next order changes).

**Excess rates and the role of variability**

Figure 3a, b displays the differences in the warming rates of *TXx* and *TNn* relative to the seasonal averages (excess rates), at the +2K GWL. As already noted, major excess rates are projected at middle and high latitudes, where the coldest nights warm faster than the nighttime mean while the hottest days warm slower than the daytime mean, in all seasons but summer. At lower latitudes instead, *TXx* show local to regional amplifications compared to *TXm* in every season, mainly over tropical lands. Hence, according to Eq. (1), changes in the higher moments as a group provide either additional boosts or delays, which may be comparable in magnitude to (or even overshoot) the changes in the averages (Supplementary Fig. 4). They substantially modify the seasonal warming patterns, exacerbating regional differences in the scaling behavior of the extremes and making cold warming rates much higher than the hot ones in many places, particularly in the winter hemispheres.

Among the higher moments, variability plays a prominent role, as evidenced by the pattern similarities between the excess rates (Fig. 3a, b) and the rates of change in standard deviation *TXsd* and *TNsd* displayed by season in Fig. 3c, d (at the +2K GWL). The latter point to rapid large-scale decreases in variability at middle and high latitudes (except for summer) and milder regional increases at lower latitudes, remarkably paralleling the excess warming of the extremes all over the globe. Correlations are large and positive in *TX* anomalies, negative in *TN* (Supplementary Table 3). Indeed, as is clear from Eq. (1), decrease in variability at mid-to-high latitudes tends to bring extreme anomalies closer to the mean, accelerating warming of the coldest nights while suppressing that of the hottest days. In contrast, the regional increases in *TX* variability force the hottest days away from the mean, amplifying their warming. In the case of *TN*, reduction of variability is prevailing in most places and concurs to the more intense warming of cold than hot extremes globally (e.g., Fig. 1b, c). Both *TXsd* and *TNsd* changes, like changes in the extremes, are approximately linear with local warming (Supplementary Fig. 5) and in many regions robust in sign across models, as discussed later on.

Figure 3e, f illustrates the zonal contributions of variability to the excess warming of *TXx* and *TNn*. In the left panels, warming rates of the extremes are compared in the zonal and multimodel mean to those of the seasonal averages, to highlight amplitude and direction of major deviations. In the right panels, by following Eq. (1), *TXx* and *TNn* excess rates are scaled by the early-industrial extreme-to-mean distance (dashed lines) to allow for a quantitative comparison with the corresponding fractional rates of change in *TXsd* and *TNsd* (black lines). Also shown here are the total spreads in model projections (shaded bands). Although uncertainties are rather large in either case, fractional rates in standard deviation well agree in the multimodel mean with the scaled excess rates in most places. In particular, they both tend to be distinctly negative at subpolar and middle latitudes (except for boreal summer), indicating that the extreme-to-mean distance narrows under warmer conditions due to decreased variability. As a result, winter warming of *TNn* is greatly amplified and, at northern mid-to-high

latitudes and over the Southern Ocean, can overshoot the seasonal background by up to 30% and 50%, respectively (Fig. 3f). Meanwhile, winter *TXx* warming is suppressed by even larger amounts (Fig. 3e). Instead, during summer the slow changes in both *TXx* and *TNn* at high latitudes mainly stem from the background seasonal warming, with small (or null) contributions from the higher moments.

As seen in Fig. 3e, in some places the fractional decreases of *TXsd* appear slightly faster than expected from the hot excess rates, revealing residual contributions from changes in the non-Gaussian properties of *TX* distributions, namely, skewness and/or kurtosis. For example, at southern latitudes *TX* skewness shows increasing tendencies that, by opposing variability decrease, may favor *TXx* warming (Supplementary Fig. 6a). With a few exceptions, non-Gaussian changes as a whole play a minor role in the zonal approximations, because of their patchy spatial structures and, perhaps, of mutual cancellations. However, their effects may be appreciable at local-to-regional scales[38,41].

Table 1 summarizes the global area-weighted contributions of the changes in the first and higher moments to the warming patterns of the extremes in every season. The contribution of variability is crucial to understanding their large regional unevenness, as proved by the fraction ($R^2_{full}$) of the total spatial variation of *TXx* and *TNn* changes which is explained if corresponding changes in *TXsd* and *TNsd* are taken into account in Eq. (1). This fraction is always much greater than that explained by the seasonal background alone ($R^2_{red}$) and can exceed 90%. The gain is larger for *TNn* than *TXx* since the latter receive regionally relevant contributions from non-Gaussian changes in *TX* anomalies, as mentioned above.

The effects of changes in variability clearly depend on regions and seasons, as shown by the zonal decomposition of both $R^2_{full}$ and $R^2_{red}$ in Fig. 4. For example, in the summer hemispheres (Fig. 4a) *TXsd* changes over the tropics, although moderate in magnitude, are at least as important as changes in the averages to explain *TXx* warming patterns, whereas they seem largely negligible elsewhere. Conversely, during winter (Fig. 4b) *TNsd* changes give the most significant contributions in a rather symmetric way at mid-to-high latitudes, where they account for the detailed spread and amplification of *TNn* warming.

**Physical insights into higher order changes**

Changes in distributions may have a profound impact on future climate, however, the mechanisms underneath have yet to be fully understood. Midlatitude variability decrease, in particular, is a robust result of models and observations[34–36,42,57–59]. Current theories attribute the underlying causes to a weakening of the meridional (equator-to-pole) gradient by Arctic amplification. This may affect thermal advection, leading to weaker anomalies and thus to a reduction of variability at the daily and longer timescales[34–36,41,42,59]. As is further evidenced by the latest simulations of a +2K climate (Supplementary Fig. 7a–f), this mechanism acts prominently during winter months, when the Arctic and the high latitudes warm faster than the rest of the globe. Correspondingly, the winter meridional gradient is seen to weaken over large swathes of land and ocean, where positive changes oppose mostly negative early-industrial gradients. Patterns of change share clear similarities with those of *TNsd* (Supplementary Fig. 7b, d), thereby corroborating previous findings[34,36,41,42]. Regional factors may also contribute, such as direct effects of sea-ice and snow cover decline[28,31,39,58] as well as changes in the zonal (east-west) gradients[36]. The latter slacken in winter with greenhouse warming (Supplementary Fig. 7e, f), as cold continents warm faster than the water masses, accelerating *TNsd* decrease off the continental coasts.

During summer (Supplementary Fig. 7g–n), horizontal gradients are smoother and, in many places, tend to reinforce at rising GWLs (the +2K changes having the same sign of the early industrial gradients), both because already warmer continents warm faster than the oceans

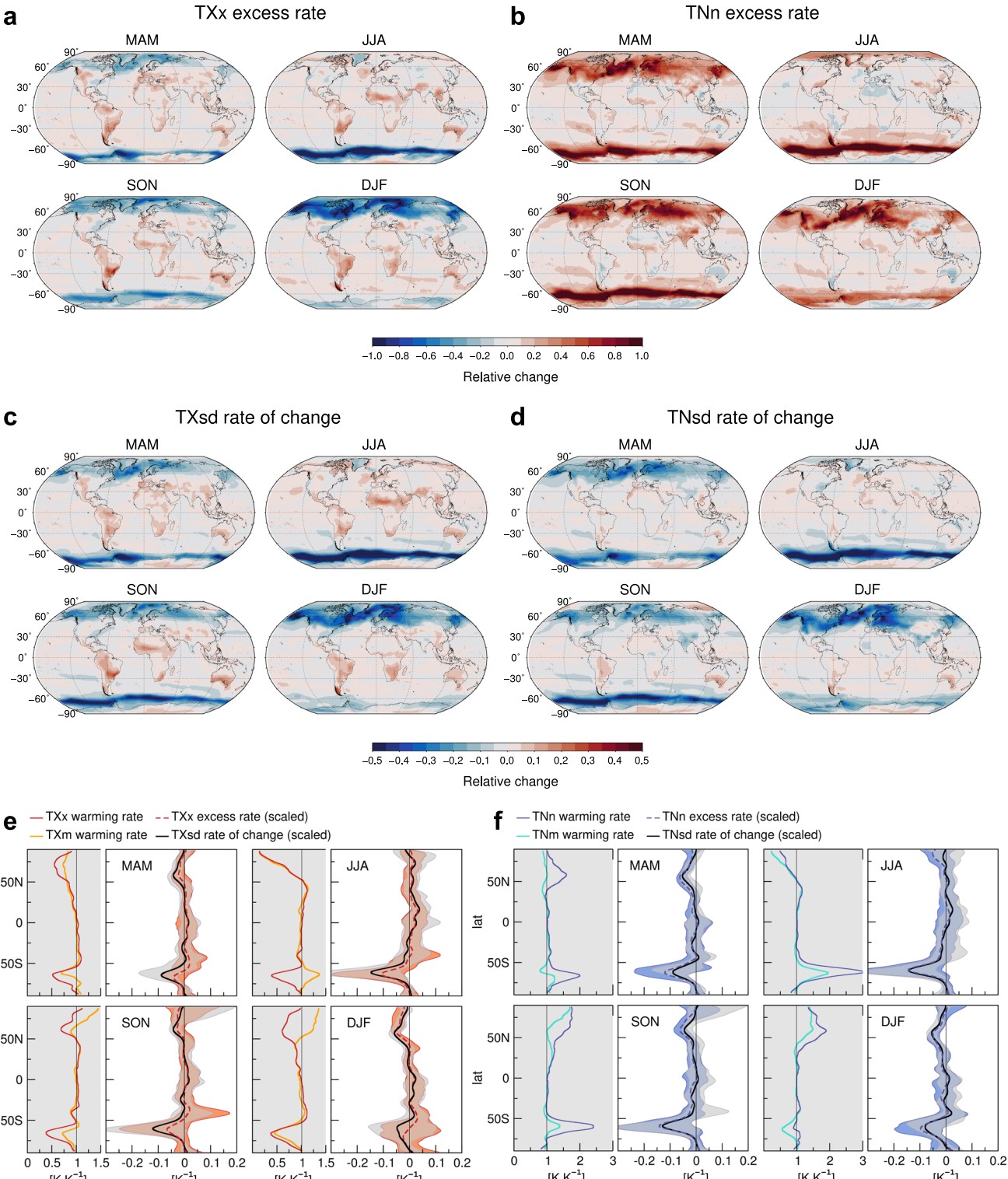

**Fig. 3 | Excess rates and higher moments. a**, **b** Excess warming rates of the hottest days (*TXx*) and the coldest nights of every season (*TNn*) respectively, that is, $(\Delta TXx - \Delta TXm)/\Delta T$ and $(\Delta TNn - \Delta TNm)/\Delta T$ where *TXm* denotes the seasonal daytime mean, *TNm* the seasonal nighttime mean and *T* the annual mean. MAM denotes March-April-May, JJA June-July-August, SON September-October-November and DJF December-January-February. **c**, **d** Seasonal rates of change in standard deviation, $\Delta TXsd/\Delta T$ and $\Delta TNsd/\Delta T$, respectively. **e**, **f** Left panels: comparison between **e** zonal-mean *TXx* and *TXm* warming rates and **f** between those of *TNn* and *TNm*. Right panels: comparison between **e** zonal-mean scaled excess rates

(dashed lines) $(\Delta TXx - \Delta TXm)/((TXx - TXm)_{EI} \Delta T)$ and corresponding fractional rates of change in standard deviation (solid black) $\Delta TXsd/(TXsd_{EI} \Delta T)$, and **f** similar for *TN*. Here *EI* denotes the early industrial averages, while in the figure legend scaling factors and zonal averaging are understood to simplify notations. Notice that both *TX* and *TN* scaled excess rates are negative if the distribution tail shortens and the extreme moves closer to the mean. Shaded bands denote the intermodel spread for the scaled excess rates (**e** light-red and **f** light-blue) and for the fractional rates in standard deviation (gray). All results relate to +2K warming.

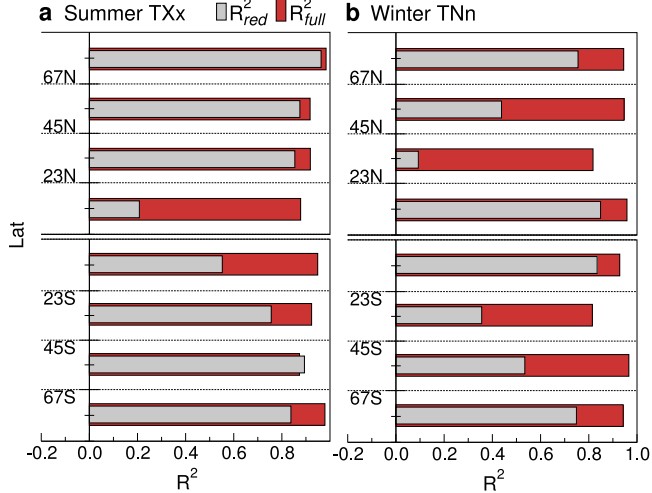

**a** Summer TXx $\quad \square R^2_{red} \quad \blacksquare R^2_{full}$ **b** Winter TNn

**Fig. 4 | Contributions of the leading moments to the extremes.** Zonal decomposition of the coefficients of determination $R^2$ of the warming rates of the hottest days (*TXx*) and the coldest nights (*TNn*) in the summer (**a**) and the winter hemispheres (**b**), respectively, at +2K warming. As in Table 1, $R^2_{red}$ accounts for changes in the averages alone whereas $R^2_{full}$ accounts for the first two moments (excluding skewness and next order changes).

and warming is non-uniform over land. Thus, stronger gradients imply stronger thermal anomalies, which amplifies variability[36,41]. Summer *TXsd* changes are rather consistent with those projected in the mean gradients (Supplementary Fig. 7h, l). For example, *TXsd* increase in northern and eastern Europe may be partly explained by a regional strengthening of the meridional gradient, due to the faster background warming of the Mediterranean zone[35,36,39,41]. Several areas, such as the Arctic coasts, northern Africa and Australia may also feel a strengthening of (either negative or positive) gradients, which nicely matches the expected *TXsd* increases. Along with advection-based mechanisms, land surface and radiative processes can additionally affect the higher moments beside mean temperatures. In particular, summer increase in variability over midlatitude and tropical regions (including, e.g., South America) has been directly related to surface drying and changes in turbulent heat fluxes by soil moisture-temperature feedbacks[26,32,33,37–39,55,56,60].

Land-atmosphere interactions may also induce skewness changes[37,38], although the latter are generally uneven and more uncertain than the lower order changes (Supplementary Fig. 6). Yet one exception is the summer increase of skewness over the Southern Ocean, which has detectable effects on *TXx* warming (Fig. 3e) and could be physically connected by nonlinear advection arguments to changes in the meridional gradient and temperature variability[40]. To date, however, only a few studies have explored skewness changes[40,41,46], whose origins thereby remain unclear.

Despite recent progress, the overall knowledge of changes in the moments beyond the mean and of the root causes is still at an early stage, and further work is needed to pinpoint pivotal processes, e.g., by way of targeted model experiments. On the other hand, since the most serious effects of higher order changes are to exacerbate extreme events, assessing to what extent such changes contribute to future hotspots remains a primary aim, as is further discussed below.

### Regional hotspots
Figure 5a–f displays the scaling behavior of the extremes and of the leading moments over major hotspots (Fig. 2). Panels a, b show total changes in *TXx* and *TXm* as a function of GSAT, regionally averaged over the Euro-Mediterranean zone (EMD) in JJA and Central South

America (CSA) in SON, respectively (land only). Over both regions *TXx* warming is much faster than over the global land (by ~30%, Fig. 5e and Supplementary Table 2) and well exceeds the regional background ($\Delta T$), obeying approximate linearity with GSAT. The robustness of these changes across models is similarly high in the local perspective, since ~80% of both the EMD and the CSA domain may experience *TXx* warming stronger than the regional background ($\Delta T$~2.5K) in a +2K future, according to at least 15 out of the 20 models (Supplementary Fig. 8a, b).

As is clear from Fig. 5a, *TXx* changes in the EMD zone substantially follow those of the summer average *TXm*, with little contribution from the higher order moments, in line with observational results[13,15,61]. Indeed, summer *TXsd* changes at the +2K GWL are rather modest (yet significant, Fig. 5e) and mainly projected over northern and eastern Europe (Supplementary Fig. 8d), although they are seen to grow with the GWL, consistently with previous CMIP5 projections of the late 21st century[21,36,39]. Also, most models predict skewness decrease over Europe[41] (Supplementary Fig. 6a), which could offset variability increase and thus explain the closeness of *TXx* and *TXm* warming trajectories throughout the simulation period. The intermodel spread in skewness projections, however, is large and contributions from next order (kurtosis) changes cannot be ruled out. By contrast, *TXsd* increase in the CSA region is statistically significant (Fig. 5e) and robust across models over much of the domain (Supplementary Fig. 8e), already at the +2K GWL. This change adds to the intense warming of the seasonal average and causes a ~10% amplification of *TXx* in the regional mean (Fig. 5b).

The rapid *TXx* warming in major continental hotspots is thus primarily supported by the strong seasonal background and likely due, as noted above, to surface drying and soil moisture-temperature feedbacks[26,27,33,37,55,56]. In particular, the varying strength of land-atmosphere coupling across models is deemed to be responsible for the broad range of *TXx* projections, and also for some discrepancies with observational results[26,62]. Circulation changes, although uncertain in future scenarios, may further enhance summer heat over midlatitudes, leading to more persistent anticyclonic patterns[23–25].

Figure 5c, d displays regional changes in winter *TNn* and *TNm* over the Pan-Arctic zone (PAr) and northern mid-to-high latitudes (MHL) respectively, including all land and ocean. The Arctic shows by far the highest sensitivity to rising levels of warming, with total *TNn* changes about three times the global ones (Fig. 5f and Supplementary Table 2), in line with previous findings[1,2,12]. In the MHL zone *TNn* changes are slower, but still highly significant and more than twice the global ones (Fig. 5d, f). At the +2K GWL, the local changes robustly exceed the background warming over most of the Arctic and midlatitudes (Supplementary Fig. 8c), reaching a maximum over areas of rapid sea-ice retreat, like the Barents-Kara seas[63,64].

In the Arctic, winter *TNn* increase is largely due to the strong seasonal warming, while the concurrent *TNsd* decrease -coming mainly from the Barents-Kara sea region- provides a supplemental boost of ~10% (Supplementary Fig. 8f). Conversely, at mid-to-high latitudes *TNsd* decrease is pervasive and robust almost everywhere, accounting for about one-third of the regional increase in *TNn* (Fig. 5d, f). The latter thereby receives a remarkable contribution from winter variability decrease, which is mainly related to the meridional gradient weakening by Arctic amplification, and is expected to continue near-linearly with growing levels of warming (Supplementary Fig. 9). Similarly, over the Southern Ocean about half of the winter increase in *TNn* (Supplementary Table 2) stems from *TNsd* decrease, which has the highest rates at these latitudes (Fig. 3f).

Finally, it should be stressed that overshooting the +2K global warming limit would trigger increasingly complex and heterogeneous changes around the world, with unprecedented shifts from present-day climate conditions and ever more disruptive extreme events. In fact, +3K of global warming may result in a ~50% higher warming of hot

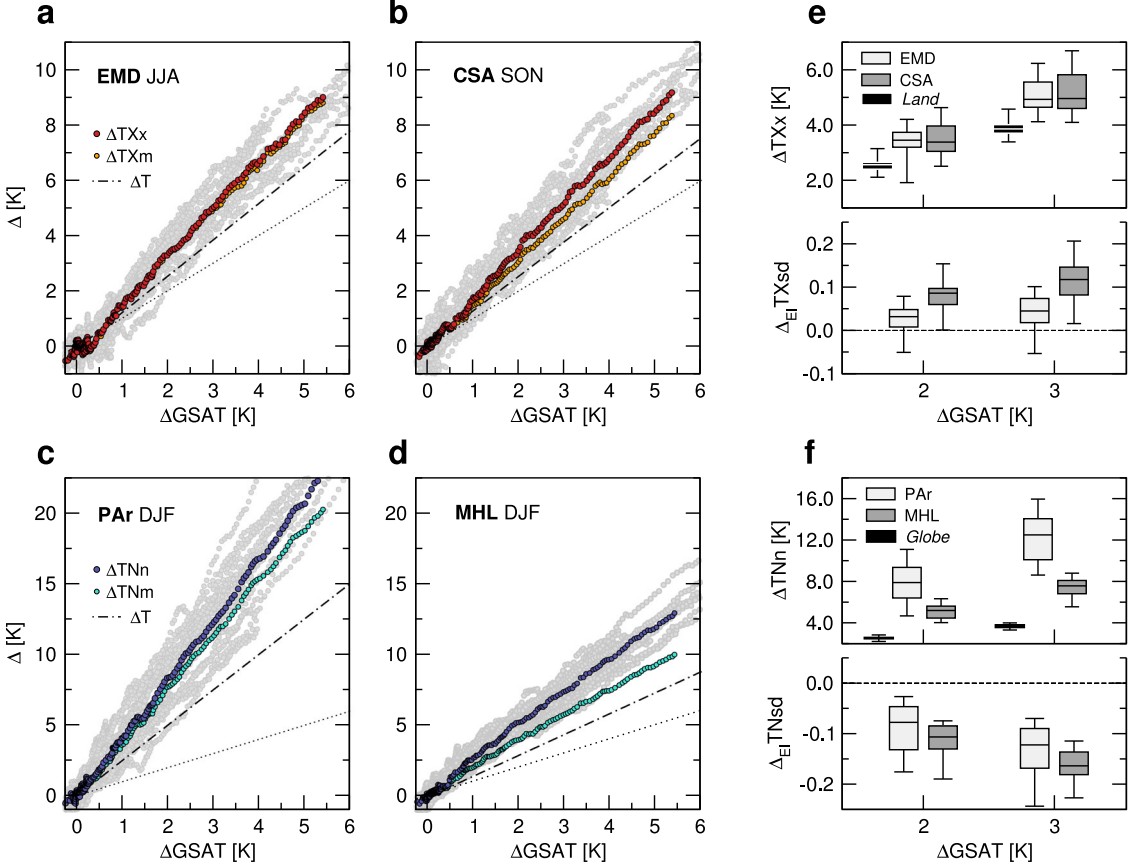

**Fig. 5 | Scaling behavior of regional extremes.** Multimodel total changes in **a**, **b** the hottest days *TXx* and **c**, **d** the coldest nights *TNn* against changes in global-mean surface air temperature (GSAT), regionally averaged over **a** the Euro-Mediterranean zone (EMD, land-only grid points in 30N-55N, 10W-40E) in June-July-August (JJA), **b** Central South America (CSA, land-only grid points in 2S-30S, 40W-73W) in September-October-November (SON), **c** the Pan-Arctic zone (PAr, 67N-90N) and **d** northern mid-to-high latitudes (MHL, 45N-67N) in December-January-February (DJF). Gray dots denote the intermodel spread. Also shown are the corresponding multimodel changes in the seasonal (Δ*TXm* and Δ*TNm*) and the annual averages (Δ*T*, given by straight-line fits). Dotted lines denote the identity relation. **e**, **f** Top panels: distribution of *TXx* and *TNn* changes across model projections at +2K and +3K warming, for the above-defined regions. Annual changes in land averaged *TXx* and global *TNn* are also shown in **e** and **f**, respectively. Bottom panels: distribution of fractional changes in standard deviation **e** *TXsd* and **f** *TNsd* across models, for the same regions and warming levels as above. Box-plots display the five-number summary statistics (as in Fig. 1a). All changes are referenced to early industrial times and based on the high-end scenario SSP5-8.5.

extremes over large continental areas and a further 4K-warming of cold extremes over the Arctic (Supplementary Table 2).

## Scaling rules of exceedance probabilities

Unusual hot events in the recent past may become ever more common under warming scenarios. Figure 6a, b shows the evolution of global and land averaged probabilities for *TX* anomalies to exceed the early-industrial 99th percentile (hot days) and the 99.9th (very hot days) respectively, as a function of changes in GSAT under alternative SSPs. The exceedance thresholds (determined for each model and grid point, see "Methods") select fixed magnitude events which occur approximately three times a year and once every three years, respectively, in the early industrial climate. As GSAT increases, the probabilities of the extremes, like their magnitudes, grow at the same rates irrespective of the forcing scenario (Fig. 6a, b). Unlike magnitudes, however, probabilities are nonlinear with GSAT and grow faster in the global than the land average due to the huge contribution from the tropical oceans. This is shown in Fig. 6c (and Supplementary Fig. 10a), where multimodel changes in hot (and very-hot) day probabilities are represented in fractional terms at the +2K GWL relative to the early industrial. Changes in probability are opposed, in some respects, to changes in magnitude, being larger over the oceans than over land and much larger in the tropics than in polar regions (despite the rapid increase in *TX* warming with latitude, Fig. 6d). As is shown in

Supplementary Fig. 11, furthermore, hot event probabilities increase disproportionately with the level of warming, the more (in fractional terms) the rarer the events.

The intrinsic nonlinearity of exceedance probabilities is critical to explaining their scaling with greenhouse warming and their highly heterogeneous changes across the globe. This can be seen in a Gaussian approximation of *TX* anomalies, having fixed variability to the early industrial value $TXsd_{EI}$ and a shifting mean *TXm* with global temperature *T*. Rates of change of hot event probabilities are then given by

$$\frac{\mathrm{d}P(x \geq x_+)}{\mathrm{d}T} = \frac{1}{\sqrt{2\pi}\, TXsd_{EI}} \frac{\mathrm{d}TXm}{\mathrm{d}T} \times \exp\left[-\left(\frac{x_+ - TXm}{\sqrt{2}\,TXsd_{EI}}\right)^2\right], \quad (2)$$

where *P* is the total probability of exceeding a fixed threshold $x_+$ (or $t = x_+/TXsd_{EI}$ in sigma units, see Supplementary Eq. (2)). Under these assumptions, rates of change rise exponentially as *TXm* increases (even linearly) with global warming and, for any fixed change in *TXm*, they decay exponentially as $TXsd_{EI}$ widens. As a result, changes in probabilities sharply depend on the native structure of the *TX* distribution (defined by variability in the Gaussian case). Namely, the shorter is the warm tail in the recent past, the faster is the rise in hot event probabilities as the distribution moves upward with global warming. This is

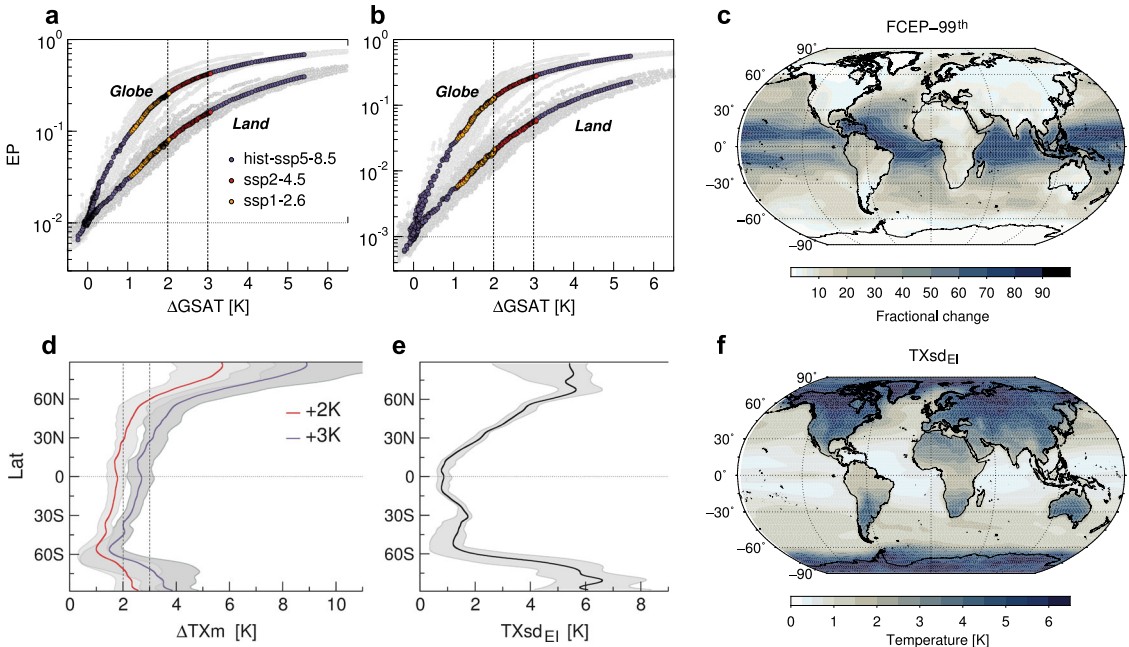

**Fig. 6 | Global changes in hot event probabilities and their key determinants. a**, **b** Evolution of multimodel global and land averaged probabilities (on a log-scale) of hot and very hot days respectively (exceedance probabilities *EP*), as a function of changes in global-mean surface air temperature (GSAT) under alternative forcing scenarios. Gray dots denote the total spread in model projections. **c** Multimodel pattern of change of hot day probabilities at +2K warming (in the high-end scenario

SSP5-8.5), expressed in fractional terms relative to the early industrial. Namely, *FCEP* = Δ*EP*/*EP$_{EI}$* with *EP$_{EI}$* = 0.01. **d** Zonal changes in the annual averages *TXm* at +2K and +3K warming. **e**, **f** Zonal and grid-point standard deviations *TXsd*, respectively, as simulated in the multimodel mean over the early industrial era. Gray-shaded bands (**d**, **e**) denote the intermodel spreads.

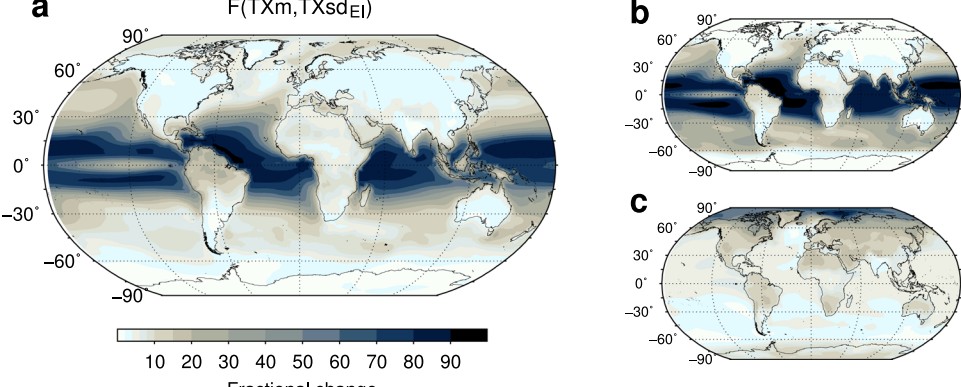

**Fig. 7 | Gaussian model for hot day probabilities. a** Theoretical probability changes obtained by integrating Eq. (2) (main text) over the +2K warming range and using multimodel grid-point results for the changes in the annual average *TXm* and the early-industrial standard deviation *TXsd$_{EI}$*. **b**, **c** Theoretical changes calculated from Eq. (2) as above, but each of the two parameters, in turn, is held fixed throughout and equal to its area-weighted global average, to disentangle the relative contributions. Respectively, **b** *TXm* = < *TXm* >$_g$ and **c** *TXsd$_{EI}$* = < *TXsd$_{EI}$* >$_g$.

The fraction $R^2$ of total spatial variation of projected changes in hot day probabilities (*FCEP*-99$^{th}$, Fig. 6c) explained by the full Gaussian model *F*(*TXm*, *TXsd$_{EI}$*) (**a**) amounts to ~0.9, and it reduces to ~0.5 if *TXm* = < *TXm* >$_g$ (**b**). Instead, if *TXsd$_{EI}$* = < *TXsd$_{EI}$* >$_g$ (**c**) $R^2$ becomes negative (~ −0.9), meaning that if the total spatial variation of *TXsd$_{EI}$* is not accounted for, the Gaussian model has worse predictions than the baseline global average of *FCEP*-99$^{th}$.

confirmed by the strong anti-correlation between projected changes in probabilities (Fig. 6c and Supplementary Fig. 10a) and native-simulated standard deviations *TXsd$_{EI}$* shown in Fig. 6e, f (Supplementary Table 4).

In fact, despite its simplicity, a Gaussian shift of *TX* distributions with fixed variability well explains the future patterns of change in exceedance probabilities, as is clear by comparing projected changes in hot days (Fig. 6c) with theoretical expectations at the +2K GWL displayed in Fig. 7a. The latter are obtained by way of Eq. (2) together with multimodel simulated *TXsd$_{EI}$* and changes in *TXm* at every grid point. Native variability combined with the shift velocity thus suffice to

predict the evolution of hot day probabilities over most of the globe ($R^2$ ~ 0.9, see "Methods"). In particular, the wide spatial variation of *TXsd$_{EI}$* is pivotal, accounting alone for the main large-scale features of projected changes in probability, such as their major amplification over the oceans and their sharp reduction from the tropics to polar regions (see Fig. 7b, c). Similar results hold for very hot days (Supplementary Fig. 10a, b) and higher levels of warming.

All this points to the leading role of native variability which, far outweighing its predicted changes as well as local *TX* warming, tightly controls future frequencies of hot and very hot days. Thus, regions showing the smallest *TXsd$_{EI}$* are those prone to the most rapid rise in

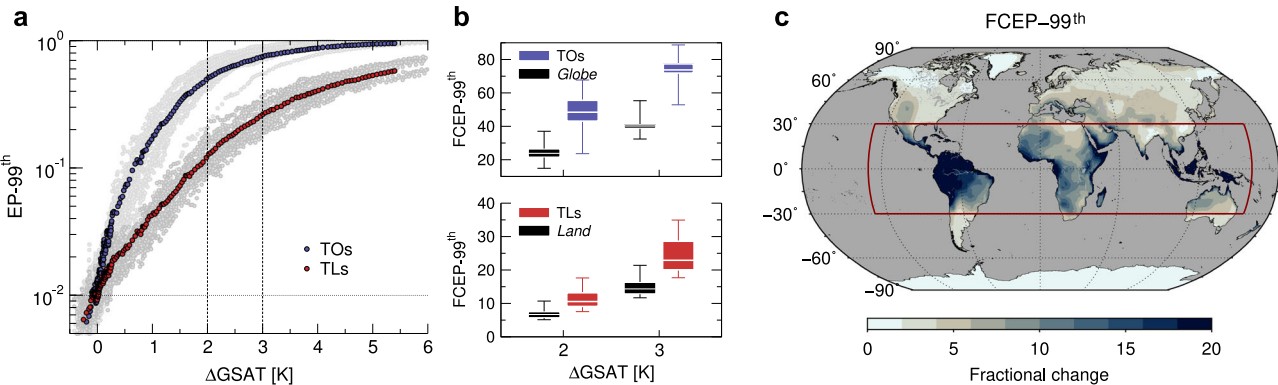

**Fig. 8 | Tropical changes in hot day probabilities. a** Multimodel hot-day probabilities (exceedance probabilities *EP*-99ᵗʰ) averaged over tropical oceans (TOs, non-land grid points in 30S-30N) and tropical lands (TLs, land-only grid points in 30S-30N), as a function of changes in global-mean surface air temperature (GSAT). Gray dots denote the intermodel spread. **b** Distributions of fractional changes in hot day probabilities (*FCEP*-99ᵗʰ) across models, averaged over TOs and the globe (top panel), and over TLs and the global land (bottom), at fixed levels of warming. **c** Multimodel fractional changes in hot day probabilities over land (ocean changes are masked) at +2K warming. All results are based on the high-end scenario SSP5-8.5.

the number of hot events. The largest increase is foreseen over tropical oceans, where native-simulated variability is at its lowest (≲1K in the regional mean). Here, despite the rather-slow *TX* warming, hot days may become commonplace already at the +2K GWL, with a ~50-fold increase relative to early industrial times, as seen by regional aggregation in Fig. 8a, b (Supplementary Table 2). At mid-to-high latitudes the larger *TXsd*_{EI}, mostly over land, counteracts the effects of stronger *TX* warming, scaling down the increase of hot events. Changes are even negligible in Antarctica, where variability is at its highest and *TX* warming relatively moderate (Fig. 6d–f).

Although significant changes are predicted in *TX* variability under greenhouse warming (e.g., Fig. 3c), they are found to contribute little to the evolution of probabilities globally. Changes in the higher moments as a whole, as well as intrinsic non-Gaussian properties are likely responsible for the fraction of the total spatial variation of probability changes (~10%) which remains unexplained by the above Gaussian model (Fig. 7a). For example, as suggested by pattern correlations (Supplementary Table 4), increases in *TXsd*, mainly projected in the tropics, may further enhance the chance of hot events regionally. Likewise, regions showing marked left-skewed distributions and/or skewness increase (like over the Southern Ocean) may experience faster rise in probability than expected by a rigid Gaussian shift.

Increase in probabilities over land, though slower than over the oceans, may still be large and significant, particularly in the tropics[3,65] as seen in Fig. 8c. Averaged over tropical lands, the hot day probability is 12-fold higher at the +2K GWL than under early industrial conditions (rising from about 3 to more than 40 events per year) and rapidly grows with GSAT (Fig. 8a, b). Likewise for very hot days, predicted to be ~34 times more frequent at the +2K GWL (Supplementary Table 2). Consistently with observed tendencies[4,66], the most impressive changes involve tropical South America (Fig. 8c) where they are enhanced by a dangerous combination of intense warm-season *TX* warming, low native variability and its increase in all seasons (Fig. 3c).

The rise in frequency of hot extremes is paralleled by a widespread decrease of cold extremes (Supplementary Fig. 12), which is especially fast again in the tropics and over the oceans due to the low *TN* variability.

Historical variability, thereby is the key to explaining the amplitude and spread of future changes in exceedance probabilities all over the globe. Furthermore, because of nonlinearity, even moderate model errors in native simulated variability can produce large uncertainties in future probabilities (Supplementary Note). This is of critical importance for regions, like the tropics, where hot events are expected to increase disproportionately.

Comparison with real data over the early industrial past can thus help evaluate projection reliability and reduce the above uncertainties. Results from a preliminary assessment are illustrated in Supplementary Fig. 13, where daily observations from the Berkeley Earth dataset[67] are used to obtain historical (1881–1910) *TXsd* over much of the global land. A glance to Fig. 6f reveals a rather poor matching between simulated and observed *TXsd* in many places[20]. Ostensibly, models tend to misrepresent historical distributions overestimating variability at subtropical and higher latitudes, whereas the opposite is observed around the equatorial belt. This means that hot event probabilities might rise faster than projected in several areas, like the EMD zone and Australia, whereas they might be slightly slower in regions such as Central America (Supplementary Fig. 13). The intermodel spread about early industrial *TXsd*, however, is rather large (Fig. 6e) and observational errors unknown. Clearly, in order to draw firm conclusions, a thorough analysis is required, including additional observational products and an estimate of their uncertainties.

## Discussion

Heat extremes are predicted to intensify rapidly with anthropogenic greenhouse warming, reaching unprecedented levels of frequency and severity in many places. Our results provide fundamental insights into the underlying amplification/damping mechanisms, showing that the high heterogeneity in future trajectories of the extremes is closely related to the fine structure of thermal distributions and their detailed evolution (Fig. 9).

Specifically, native variability is found to be crucial to explaining and predicting the global patterns of changes in the frequency of unusual hot events. It reshapes the strength of the background regional warming and, in many areas, is by far predominant. This is the case of the tropics where the intrinsically low variability causes hot events to hugely increase in number, notwithstanding the rather slow background warming. Viceversa, at higher latitudes the larger variability dampens the effects of the more intense background. Historical climate conditions are thus more important than their changes from a predictive standpoint.

How realistically climate models may represent historical distributions, therefore, has a substantial impact on projected changes in hot event probabilities, affecting both their amplitude and uncertainty. Indeed, because of the high sensitivity of probability changes to native variability, even little overestimation of the latter can lead to large underestimation of the future frequency of hot events, contributing to inflate projection uncertainty. This stresses the importance of scrutinizing models' performance in simulating true historical variability, so as to better constrain scenario projections and to narrow the

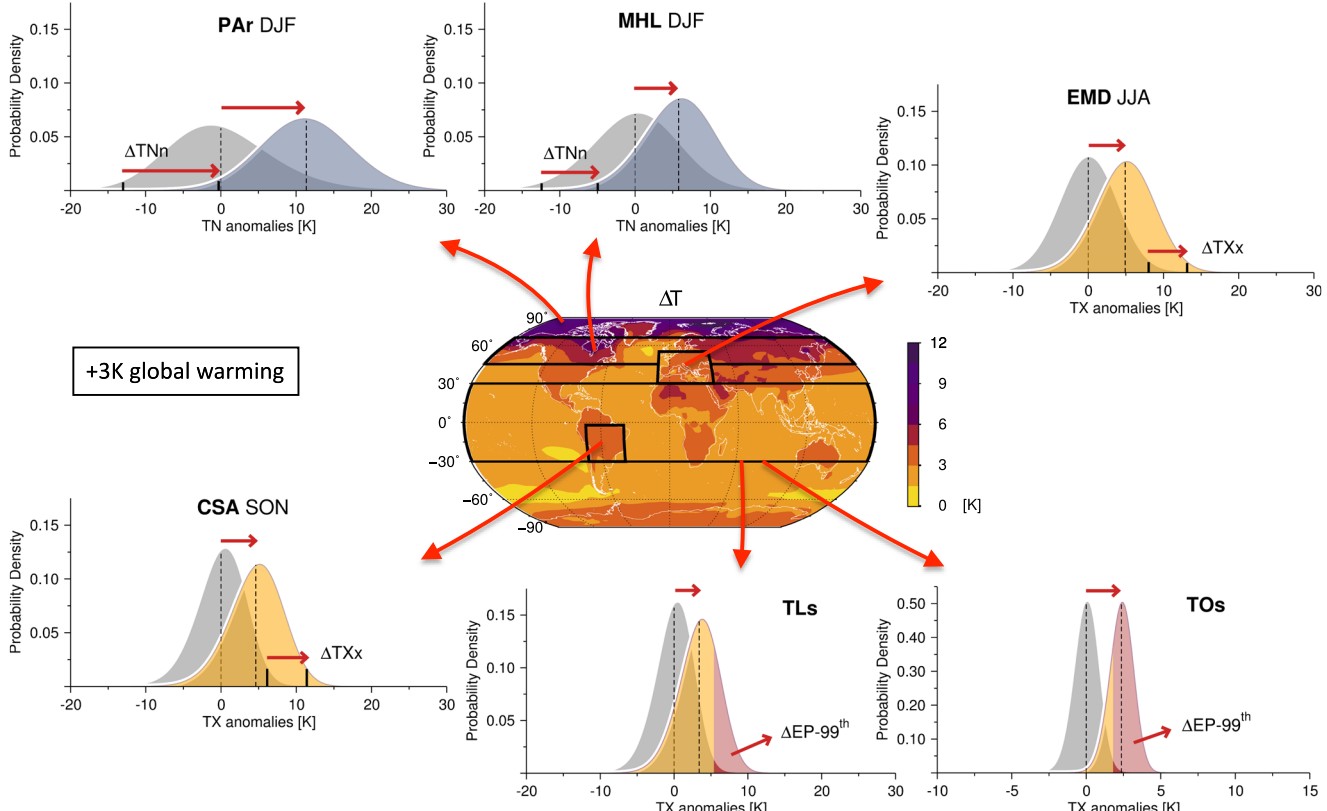

**Fig. 9 | Summary chart of major hotspot changes.** Schematic illustration of future changes in the magnitude and frequency of heat extremes over the hotspot zones at +3K warming (Supplementary Table 2), and their relations with changes in the underlying thermal distributions. A theoretical density function allowing skewness (where appropriate) is used to represent the data[47,61]. $\Delta TXx$ and $\Delta TNn$ denote the change in the hottest days and the coldest nights respectively, $\Delta EP$-99th the change in hot day probabilities and $\Delta T$ the change in annual mean temperatures.

uncertainties. This aim can be achieved either by way of weighting (or selection) algorithms which prioritize results from the best performing models[62,68] or by applying bias correction to simulated distributions such that these match the observed ones improving projection accuracy[69]. The applicability of these approaches, though, is challenged by the unequal global data coverage and remains confined to regions where reliable observations exist and go back to early industrial times.

On the other hand, future changes in variability (regardless of the historical values) are shown to deeply affect warming rates of the extremes, leading to striking deviations from the regional and seasonal background and a large unevenness across the globe. Decrease in midlatitude variability, above all, causes cold extremes to warm much faster than hot extremes in many places.

In either case, next order moments (e.g., skewness) and their changes are found to play a minor role at the global scale, however, they can be of importance regionally and need to be investigated further.

Unfortunately, there is not a direct link between the past behavior of thermal distributions and the future severity of the extremes, since their warming degree depends on the coincident changes in the leading moments. However, misrepresentations of historical changes in distributions can signal models' deficiencies in simulating relevant physical processes[58,62,68], undermining confidence on future magnitudes of heat extremes. Furthermore, how distribution moments are predicted to change under warming scenarios could help uncover the main physical drivers. A case in point is winter variability decrease at midlatitudes, which is spatially consistent with the meridional gradient weakening and points thereby to polar amplification as a key mechanism behind the prominent warming of cold extremes[34,42].

Hence, the significance of the higher statistical moments for extreme climate trajectories cannot be overstated. Their tight connections are essential to understand the differential regional response of the extremes to greenhouse warming. At the same time, they can be leveraged to improve model projections of future changes, which is a pressing goal in order to anticipate the worst consequences of climate change and to cope with them.

## Methods
### Model outputs
This study is based on a global daily-scale set of near-surface (2m) maximum (*TX*) and minimum temperature data (*TN*), taken from a multimodel ensemble of historical (1850–2014) and 21st century CMIP6 simulations[49]. A range of alternative forcing scenarios is investigated (SSP1-2.6, SSP2-4.5 and SSP5-8.5), retaining one realization per model as detailed in Supplementary Table 1.

Annual mean temperatures were first derived from monthly data throughout the simulation period, and then averaged all over the globe and across models to obtain GSAT and its changes since early industrial times (1851–1900). The timings for GSAT to exceed +2K and +3K above the early industrial mean were estimated for each model in the high-end scenario (SSP5-8.5), using 21-year running averages. Crossing periods were defined by model-specific 21-year windows around the exceedance years (listed in Supplementary Table 1) and then routinely used to compute the expected values of relevant observables at the given level of warming.

### Daily anomalies
*TX* and *TN* daily data were turned into anomalies by computing deviations from the respective day-of-the-year early-industrial normals. *TX* and *TN* daily normals were formerly estimated for every point

of the model native grids by taking the 1851–1900 mean of the absolute temperatures within a time window centered around each calendar day and then using discrete Fourier transforms to filter out noise. Upon removal of the annual cycle, daily anomalies in the early industrial era have zero mean by construction, but the detailed shape of distributions is model and grid point specific at any time.

### Statistical moments

The leading properties of grid-point *TX* and *TN* distributions (mean, variability and skewness) were assessed for every single year and season throughout the simulation period and for each model and scenario, using the first and the higher-order central moments. These read

$$\mu_1 = E[x] \quad \text{and} \quad \mu_k = E[(x - \mu_1)^k], k = 2,3 \qquad (3)$$

respectively, where $E[\ ]$ is the expectation operator and $x$ the temperature anomaly. Distribution standard deviation is given by the square root of the second moment about the mean, $\sigma = \sqrt{\mu_2}$. Skewness is the third standardized moment $\gamma_1 = \mu_3/(\mu_2)^{3/2}$ and measures the degree of asymmetry between cold and warm anomalies. For example, skewness is negative if the tail at the left end of the distribution is longer than the tail at the right end, meaning that very cold events are more likely to occur than very hot ones.

Empirical moments were computed from Eq. (3), using model grid-point data and common sample estimators[70]. Specifically, the latter were derived for every single year from annual and seasonal samples of daily anomalies (at least 360 and 90, respectively). The choice of the data samples is aimed at removing spurious trends in the higher moments induced by the increasing mean (which is subtracted year by year, Eq. (3)) and, furthermore, at tracing the time evolution of the daily distributions from which the extremes are drawn.

### Heat extremes

Magnitudes and probabilities of the extremes were obtained, as distribution moments, for each model on its native grid. Magnitudes were analyzed for the fixed-probability events given by the hottest daytime (*TXx*) and the coldest nighttime anomaly (*TNn*) of every year and season in 1851–2100, under alternative forcing scenarios. Probabilities of fixed-magnitude extremes were defined using, throughout the simulation period, climatological exceedance thresholds (i.e., the 1st, the 99th and the 99.9th percentile) drawn from the grid-point anomaly distributions of all days of the year in the early industrial era (1851–1900, disregarding seasons).

Local changes in statistical moments and the extremes under alternative forcing scenarios were referenced as a rule to the early industrial averages. Global and regional area-weighted aggregations were then performed for each model using native grid results, and finally re-expressed as a function of projected changes in GSAT.

Uncertainty in future projections is assessed to a first approximation by the total spread across models, since this tends to be broader than, and to encompass, individual intramodel spreads, as already noted in previous studies[31,36]. This is further shown in Supplementary Fig. 9, where the intermodel and intramodel spreads are compared, as a matter of example, in the case of midlatitude changes in *TN* variability, by exploiting the run multiplicity of a couple of GCMs.

### Multimodel patterns of change

Global patterns of change in distribution moments and the extremes, as projected by each model at the +2K and +3K GWL, were estimated by taking at each grid point the time average over the model-specific 21-year period associated with the given increase in GSAT, as stated above (*Model Outputs*). Native grid-point changes, always referenced to the early-industrial averages, were later remapped to a common coarse-grained grid (Gaussian N32), before averaging across models and over the latitude circles.

The agreement between model simulated patterns of change in the magnitude and frequency of the extremes and the theoretical expectations based on Eqs. (1) and (2), respectively, was quantified by the coefficient of determination

$$R^2 = 1 - \frac{SS_{\text{res}}}{SS_{\text{tot}}}; \qquad (4a)$$

$$SS_{\text{tot}} = \sum_i w_i (y_i - \bar{y})^2, \quad \bar{y} = \sum_i w_i y_i \qquad (4b)$$

$$SS_{\text{res}} = \sum_i w_i (y_i - \hat{y}_i)^2, \qquad (4c)$$

where $y_i$ is the multimodel grid-point change at the given GWL ($i$ running over the total number of points), $\hat{y}_i$ its theoretical counterpart and $w_i$ the grid point weight (cell area fraction). In the case, e.g., of *TXx* warming rates (Table 1), $y_i$ is the grid-point seasonal change per degree of local warming $\Delta TXx/\Delta T$ at the +2K GWL, while $\hat{y}_i$ is given by the corresponding distribution changes combined according to Eq. (1). Likewise, for probabilities (Fig. 7), $y_i$ is the grid-point fractional change in hot day probabilities at the +2K GWL, while $\hat{y}_i$ is its analog obtained from Eq. (2).

The total variation $SS_{\text{tot}}$ of model changes about their global mean $\bar{y}$ gives the degree of their spatial heterogeneity, and $R^2$ is a measure of how this is captured by the theoretical expectations. Namely, the smaller is the area-weighted sum $SS_{\text{res}}$ of squared residuals, the closer to 1 is $R^2$, i.e., the larger is the fraction of $SS_{\text{tot}}$ explained by theoretical changes. Viceversa, if the latter have worse predictions than the global mean $\bar{y}$, then $R^2$ becomes negative.

### Data availability

The model data used in this study are publicly available at https://pcmdi.llnl.gov/CMIP6/. The Berkely Earth observational dataset is available at http://berkeleyearth.org/data/. Source data used for the figures are provided with this paper.

### Code availability

Computer codes developed for data analyses are available on request from the authors.

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

## Acknowledgements

This work has received funding from the Italian Ministry of Education, University and Research (MIUR) through the JPI Oceans and JPI Climate "Next Generation Climate Science in Europe for Oceans"–ROADMAP Project (D. M. 593/2016) (S.C.), and from the European Union's Horizon 2020 research and innovation program under grant agreement No. 820970 (TiPES) (S.C.). This is TiPES contribution # 179. The authors thank the climate modeling groups for producing and making available their model output, and the Earth System Grid Federation (ESGF) for archiving the data and providing access.

## Author contributions

C.S. and S.C. designed the research; C.S. performed calculations; C.S. and S.C. discussed the results and wrote the paper.

## Competing interests

The authors declare no competing interests.
