## [Peer Review File · Nature Communications]

Quantifying the role of variability in future intensification of heat extremesREVIEWER COMMENTS

Reviewer #1 (Remarks to the Author):

This paper is a very interesting contribution to the field of explaining changes in temperature extremes by means of focusing on the statistical properties of the temperature distributions. In this lies also my main concern: the paper basically only addresses statistics. Very little physical reasoning is provided to support the statistical findings, aside from a few instances where the paper cites earlier work. I would strongly urge the authors to add information on what processes are behind the changes in the mean, the variability and certain statistics e.g. skewness. Otherwise the paper is merely an exercise in statistics. A very interesting one, but without physics it is incomplete. For instance, I can imagine that certain changes in the statistical properties might point at certain processes being pivotal, e.g. for increases in Tx soil drying or circulation changes, among other processes. This would make the paper much more interesting and important.

One specific comment: the paper seems to conclude that SD is an important factor in relation to changes in T-extremes, but if I understand their method section correctly the paper uses only one member per model, thus disregarding a wealth of information on model variability for models that have more than one member. I do not understand why this paper does not make use of all available model members to address variability.

Reviewer #2 (Remarks to the Author):

Review of "Quantifying the role of variability in future intensification of heat extremes" submitted for publication in Nature Communications.

Key results:

This study investigates the differential nature of the global and regional future changes in daily heat extremes, namely TXx and TNn, focusing on the leading moments, namely seasonal mean and variability, of the thermal distribution. Utilising the latest historical and future scenario simulations from the CMIP6 archive, the authors provide robust model evidence that historical temperature variability is a good predictor for explaining changes in the frequency of hot extremes. The authors also show that future changes in variability are found to significantly impact the rate of warming of the temperature extremes, as shown for Eurasia in the boreal winter. For regions like South America and southern Australia, increased variability acts to enhance the warming. This topic is highly relevant given the world is likely to reach and exceed 2K warming before mid-century.

Validity:

I feel the authors have provided a clear and robust interpretation of the model data. Despite there being a substantial amount of information to digest in the 8 Figures and 11 Supplementary Figures (e.g., Figure 3 alone is quite detailed), I feel the authors give the right amount of detail in the figure captions to understand what is being. Figure 3 is a good example of that. It is a shame that observations cannot be used to validate the models, as noted in L49. I would have liked to get some idea of the climate model representation of observed distribution, even if only for a few regions, yet I except that ocean surface observations (of extremes) going back prior to the early 20th century are sparse.

Significance:

This is a hugely significant study, nicely balancing a global perspective on temperature extremes with a critical regional analysis, focusing on South America, polar regions and Euro-Mediterranean. It draws attention to the important role of variability in explaining

the TXx and TNn warming rates, particularly in the tropics in summer (TXx) and mid-latitudes in the in winter (TNn) – referring to Figure 4. I think the regional hotspot analysis is particularly compelling, comparing regional changes over South America, where variability increases amplify hot extremes in their spring, to Europe-Mediterranean, where hot summer extreme changes are due to the season mean warming.

Data and methodology:

The CMIP6 data used in this study is well documented and used in many studies, many of which are cited. The process whereby the authors assess the models is sufficiently described in the Methods section, making this study easily repeatable. The figure captions provide sufficient detail to follow what is being shown, so the authors should be commended for this.

Clarity and context:

The manuscript is quite clear and easy to read. At times, I did feel as if there was a strong reliance on the supplementary material, and this does partly break the flow of the paper. Previous work has been acknowledged and cited appropriately in the manuscript.

Suggested improvements:

I have very few improvements to suggest, as I feel the paper is already in good shape. The figures are well produced, colour schemes easy to read, and the interpretation of the results is thorough. I suggest formal publication after these minor changes are made.

→ Figure 3 is very detailed, so I wonder if the authors would consider either splitting this up into two figures (excess rate maps in one and zonal-mean plots in another) or putting the zonal-mean plots into the supplementary. It is quite difficult to distinguish the lines in (E) and (F), particularly when viewing the plot (on an A4) is quite small.

→ Figure 4: The assumption here is that $R^{>2}$ full excludes higher than second order moments. Is that correct?

→ L285: Perhaps I missed this somewhere, but what is the cause for the Tx skewness increase in the Southern Ocean in summer, referring to Suppl. Fig S6A? Also referring to Fig. S6A, the authors state (in L365-366) that for the Europe-Mediterranean region, "local variability increases may be offset by coincident decreases in skewness". The change in summer skewness over EMD looks to be between -0.05 and 0. What is the distribution like in the model skewness, and how does it compare to the fractional change distribution in Figure 5F?

→ The authors might like to consider a summary figure/schematic that succinctly details their main results, splitting this into TX and TN, with particular focus on the hotspots.

→ L20-21: "hugely increase in number". Is this the same for duration or only frequency?

→ L24: This will also affect ecosystems as well as human beings (i.e., fauna that cannot thermoregulate in hot conditions).

→ L101: This might sound better as "before the middle of the century".

→ L139-140: Suggest change to "extratropical land regions".

→ L159: This is the case for boreal summer, but for austral summer, there appears to be a warming along the Antarctic coastline in DJF. Is this related to sea-ice melt?

→ L531: I'm not sure what is meant by "its fast increase year-round". Are the authors referring to annual mean warming?

→ L562: Suggest "human-induced" or "anthropogenic" instead of "man-made".

→ L646: Interested in understanding what the justification is for using a 21-year running average to determine the GSAT exceedances? Is this standard practice? Also, is the exceedance year the middle year in question? For example, for ACCESS-CM2 passing 2K in 2038, is this the 2028-2048 average?

→ Table S4: Is the assumption that all these correlations are statistically significant?

References:

The authors have provided a suitable list of prior research.

Reviewer #3 (Remarks to the Author):

First review of 'Quantifying the role of variability in future intensification of heat extremes', submitted to Nature Communications

* General comments *

This paper uses the current generation of climate projections (CMIP6) to investigate how both the historical temperature variability and its projected change contribute to the changes in temperature extremes. It is argued that (i) in regions where the historical temperature variability is small (typically the tropics), the future warming will result in more frequent exceedances of present-day high percentiles than elsewhere and that (ii) in regions/seasons where the temperature variability increases, the hot (cold) extremes will warm faster (slower) than the mean, and vice-versa. Such statistical behaviors are qualitatively well known, but here the authors provide several ways to *quantify* them, which seems original to me, and is definitely of high interest to the scientific community.

The paper is well written, structured and illustrated, and the supplementary information is helpful to readers interested in more details. Figures are of good quality and contain a lot of information (many panels) --- I would however suggest to enlarge the maps in Figs 1, 2 and 3 which are currently hard to read on a printed version.

Overall I really enjoyed reading this paper and I am convinced that it will be of high interest for readers of Nature Comms. However I missed some important methodological details so that I could not fully understand several parts of the paper. In particular in the text associated with Fig 3 and Table 1, I did not fully understand how TXsd is calculated, so that I could not follow the discussion and be convinced by the conclusions.

Therefore I am afraid that I can not recommend publication at this stage, because in my opinion the methodology should be better explained so that the readers can understand, and in fact reproduce, the analysis made by the authors. Below are some more specific comments and suggestions.

* Specific comments *

- Fig 2 and associated text. The abbreviations TXx or TNn are traditionally used for annual max/min of daily temperatures in the climate community. I would suggest that the authors better explicit that they use the same abbreviations for seasonal max/min to avoid any confusion. Also, how does the annual cycle affect seasonal TXx or TNn? For instance, in the boreal spring, I would expect the maximum TX in MAM to often occur in late May due to the annual cycle, and I wonder how this kind of features can affect the

analysis.

- Eq 1 and throughout the text. It should be further explained how TXsd is computed, i.e. which exact sample is used to compute the standard deviation. Is it the sample of the N years (1851-1900 or the '+2K period') x 90 days (a season) of daily temperature anomalies? Is it detrended so that the long-term warming does not artificially increase the variance?

- Table 1. It is unclear to me how Eq 4 translates into the values reported here, and how these values are sensitive to the horizontal resolution --- the spatial variance would certainly change for another resolution, but would it affect the relative fractions?

- Fig 3. There are many acronyms here (and in the first part of the paper) and maybe the most complicated ones such as 'Delta_T TX x m' or 'Delta_EI,T TX x m' could be avoided.

- lines 322-328. Note that all these features are not independent (e.g. the Arctic sea-ice retreat and the reduction in the meridional thermal gradient).

- lines 332-337. Note that also zonal thermal gradients are affected under climate change (e.g. because land warms faster than ocean), and that it can also contribute to change the temperature variability at a given place. For instance, in Europe, the decrease in winter temperature variability is also associated with a smaller contrast between the climatologically mild ocean and cold continent.

- lines 362-365. Surprising because there is a large body of literature (e.g. articles 40, 64 or 65 listed in the references) that suggests an increase in European summer temperature variability due to land-atmosphere feedbacks. Has this increase disappeared in CMIP6?

- line 451. Does the 't' actually mean something in the definition of x_t (e.g. a number of sd) or is it just a typo?

- lines 602-605. It seems to me that bias correction techniques could easily deal with the issue discussed here, in particular methods that rescale the full pdf/cdf of the historical period. The authors might consider discussing the use of such techniques here.

Answer to the Reviewers

Manuscript NCOMMS-22-13201, *Quantifying the role of variability in future intensification of heat extremes*

We thank the reviewers for carefully reading our manuscript and for their constructive comments. We have modified the manuscript to address the main issues they raised and, in particular, we have included (i) an additional section focused on the physical mechanisms behind changes in thermal distributions and (ii) a global-scale and regional comparison of models' representation of historical distributions with observations. Furthermore, we have explained more thoroughly the methodology used for our analyses.

Below we answer separately on a point-by-point basis to the reviewers' criticisms and describe in detail the changes we have made (shown with color highlighting in the manuscript).

Reviewer #1

*This paper is a very interesting contribution to the field of explaining changes in temperature extremes by means of focusing on the statistical properties of the temperature distributions. In this lies also my main concern: the paper basically only addresses statistics. Very little physical reasoning is provided to support the statistical findings, aside from a few instances where the paper cites earlier work. **I would strongly urge the authors to add information on what processes are behind the changes in the mean, the variability and certain statistics e.g. skewness.** Otherwise the paper is merely an exercise in statistics. A very interesting one, but without physics it is incomplete. For instance, I can imagine that certain changes in the statistical properties might point at certain processes being pivotal, e.g. for increases in Tx soil drying or circulation changes, among other processes. This would make the paper much more interesting and important.*

R1.1 We have expanded the original manuscript in several parts so as to provide the reader with an overall insight into current understanding of the mechanisms behind changes in distribution moments.

Specifically, we have added **lines 198-214** in section *Differential warming of heat extremes*, where we emphasize relevant aspects of seasonal mean changes, their major hotspots and plausible drivers. Furthermore, in place of previous lines 318-337, we have included an **additional section** (entitled *Physical insights into higher order changes*, from **line 340 to 412**) entirely dedicated to the likely causes of future changes in the higher moments. Here we review the main theories on the physical roots of large-scale and regional changes in variability, during both winter and summer, in light of the latest CMIP6 simulations of a +2K climate. We have thereby enlarged **Supplementary Fig.7** (*Supplementary Information*) to illustrate additional results. In this context, we also give some clues to the origin of skewness changes, although the latter are found to play a minor role at the global scale and, to date, have received little attention from the climate science community.

We have applied in the same vein a few changes to the *Hotspot* section (**lines 455-464 and 486-489**), to recall the most likely processes behind regional changes in distribution moments and extremes.

*One specific comment: the paper seems to conclude that SD is an important factor in relation to changes in T-extremes, but if I understand their method section correctly the paper uses only one member per model, thus disregarding a wealth of information on model variability for models that have more than one member. I do not understand **why this paper does not make use of all available model members to address variability.***

R1.2 The rationale for the use of one realization per model is that of assigning the same weight to all models, which come with different run multiplicities. At the same time, this approach allows us to strike a balance between the computational complexity of global daily-scale analyses and the accuracy of results. The total spread across models indeed

provides a reliable first-order estimate of projection uncertainties, because this spread generally turns out to be broad enough to encompass the spreads across different realizations of the same model, as already noted in previous studies (e.g. refs.31,36, revised manuscript). We did investigate further this issue, by relying on multiple realizations of two models and focusing on the relevant case of midlatitude changes in TN variability. Results have been reported in the **Supplementary Fig.9** (*Supplementary Information*) and recalled in *Methods* (**lines 807-814**) and confirm previous findings. That is, structural differences among climate models appear larger than internal variability within the models, likely causing much of projection uncertainty. We thus expect that additionally accounting for the intramodel spreads (with appropriate weighting factors) in cross-model averages would not change results substantially.

Reviewer #2

Review of "Quantifying the role of variability in future intensification of heat extremes" submitted for publication in Nature Communications.

Key results: This study investigates the differential nature of the global and regional future changes in daily heat extremes, namely TX_x and TN_n , focusing on the leading moments, namely seasonal mean and variability, of the thermal distribution. Utilising the latest historical and future scenario simulations from the CMIP6 archive, the authors provide robust model evidence that historical temperature variability is a good predictor for explaining changes in the frequency of hot extremes. The authors also show that future changes in variability are found to significantly impact the rate of warming of the temperature extremes, as shown for Eurasia in the boreal winter. For regions like South America and southern Australia, increased variability acts to enhance the warming. This topic is highly relevant given the world is likely to reach and exceed 2K warming before mid-century.

*Validity: I feel the authors have provided a clear and robust interpretation of the model data. Despite there being a substantial amount of information to digest in the 8 Figures and 11 Supplementary Figures (e.g., Figure 3 alone is quite detailed), I feel the authors give the right amount of detail in the figure captions to understand what is being. Figure 3 is a good example of that. It is a shame that observations cannot be used to validate the models, as noted in L49. **I would have liked to get some idea of the climate model representation of observed distribution, even if only for a few regions**, yet I except that ocean surface observations (of extremes) going back prior to the early 20th century are sparse.*

R2.1 In the end of section *Scaling rules of exceedance probabilities* (**lines 634-655**) and in **Supplementary Fig.13** (*Supplementary Information*) we discuss results of a global-scale and regional comparison of model-simulated against true historical distributions, based on the Berkeley Earth observational dataset (going back to 1880). In particular, using daily data over land, we compare model and observed TX variability (TX_{sd}) in the early industrial era, given its overwhelming importance to explaining future trajectories of hot event probabilities. We further show, as a matter of example, historical probability density functions of daily TX anomalies from both models and observations for some of the most jeopardized regions under climate change. Clearly, a thorough evaluation of models against real data, and their use to constrain scenario projections, should rely on additional observational products and account for both model and observational uncertainty, which is beyond the scope of this study and is left for future work.

Significance: This is a hugely significant study, nicely balancing a global perspective on temperature extremes with a critical regional analysis, focusing on South America, polar regions and Euro-Mediterranean. It draws attention to the important role of variability in explaining the TX_x and TN_n warming rates, particularly in the tropics in summer (TX_x) and mid-latitudes in the in winter (TN_n) – referring to Figure 4. I think the regional hotspot analysis is particularly compelling, comparing regional changes over South America, where

variability increases amplify hot extremes in their spring, to Europe-Mediterranean, where hot summer extreme changes are due to the season mean warming.

Data and methodology: The CMIP6 data used in this study is well documented and used in many studies, many of which are cited. The process whereby the authors assess the models is sufficiently described in the Methods section, making this study easily repeatable. The figure captions provide sufficient detail to follow what is being shown, so the authors should be commended for this.

Clarity and context: The manuscript is quite clear and easy to read. At times, I did feel as if there was a strong reliance on the supplementary material, and this does partly break the flow of the paper. Previous work has been acknowledged and cited appropriately in the manuscript.

Suggested improvements: I have very few improvements to suggest, as I feel the paper is already in good shape. The figures are well produced, colour schemes easy to read, and the interpretation of the results is thorough. I suggest formal publication after these minor changes are made.

→ **Figure 3 is very detailed, so I wonder if the authors would consider either splitting this up into two figures** (excess rate maps in one and zonal-mean plots in another) or putting the zonal-mean plots into the supplementary. It is quite difficult to distinguish the lines in (E) and (F), particularly when viewing the plot (on an A4) is quite small.

R2.2 We have enlarged Figure 3 as much as possible (which is now text-wide), slightly changed the color palettes to better highlight results and moved the legends outside the graphs in panels e,f. Notice that we have simplified notations as required by reviewer #3 (point **R3.1**). Also, we have enhanced the figure resolution such that details can be magnified as desired in the pdf file. Unfortunately, we could not split up Fig.3 into two figures because the limit of display items is already reached by including the new summary Fig.9 in the *Discussion* (see point **R2.5** below).

→ **Figure 4:** The assumption here is that R^2 full excludes higher than second order moments. Is that correct?

R2.3 Yes, the R^2_{full} 's in Figure 4 represent a zonal partitioning of the global ones given in Table 1 and, like the latter, do not account for skewness and next order changes, as is now explicitly stated in the **figure caption**.

→ **L285:** Perhaps I missed this somewhere, but what is the cause for the Tx skewness increase in the Southern Ocean in summer, referring to Suppl. Fig S6A? Also referring to Fig. S6A, the authors state (in **L365-366**) that for the Europe-Mediterranean region, "local variability increases may be offset by coincident decreases in skewness". The change in summer skewness over EMD looks to be between -0.05 and 0. What is the distribution like in the model skewness, and how does it compare to the fractional change distribution in Figure 5F?

R2.4 We give some clues to potential mechanisms behind skewness changes, along with variability, in the newly dedicated section (**lines 393-403**), as described in point **R1.1** above. In particular, as recalled in **lines 396-401**, the summer increase in skewness over the Southern Ocean has been explained in a recent study (ref.40 of the revised manuscript) by dynamic, advection-based arguments. Though, skewness changes have received very little attention to date, which leaves their origin still unclear.

With regards to the Euro-Mediterranean zone, the grid point changes in summer skewness displayed in Supplementary Fig.6a roughly range from -0.05 to 0 per degree of local warming (at the +2K GWL) and correspond to a total regional-mean change of approximately -0.05. We show this in the figure below (left), where the EMD skewness decrease is displayed both in the distribution across models at the +2K GWL (boxplot) and as a function of

GSAT increase. Despite being quite small, skewness decrease can be guessed, at least qualitatively, to partly explain the slight mismatch between TX_{sd} increase and the smaller change in the extreme-to-mean distance $TX_x - TX_m$, as can be seen in the figure below (right). Model uncertainty however is large and, at this stage, the detailed skewness contribution to TX_x warming cannot be disentangled from that of next order (e.g., kurtosis) changes, which requires an in-depth regional-scale investigation. The role of skewness in the EMD zone has been discussed a little more extensively in **lines 439-446** of the revised manuscript.

→ The authors might like to consider a **summary figure/schematic** that succinctly details their main results, splitting this into TX and TN, with particular focus on the hotspots.

R2.5 We have included in the *Discussion* a schematic graph (**Figure 9**) that pictorially summarizes future changes in the magnitude and frequency of the extremes over major hotspots, and their relations with changes in the underlying thermal distributions.

→ **L20-21**: "hugely increase in number". Is this the same for duration or only frequency?

R2.6 Duration of heat episodes, among other aspects, has also been increasing in several places with greenhouse warming (e.g., ref.9), however we only stressed in the *Introduction* the most striking global changes which are the focus of this work.

→ **L24**: This will also affect ecosystems as well as human beings (i.e., fauna that cannot thermoregulate in hot conditions).

R2.7 We have evidenced this aspect (**line 24**).

→ **L101**: This might sound better as "before the middle of the century".

R2.8 The change has been made (**line 102**).

→ **L139-140**: Suggest change to "extratropical land regions".

R2.9 Ditto (**line 139**).

→ **L159**: This is the case for boreal summer, but for austral summer, there appears to be a warming along the Antarctic coastline in DJF. Is this related to sea-ice melt?

R2.10 It is conceivable that there could be a direct link between ice loss and summer near-surface warming around the Antarctic margins (e.g., through ice-albedo feedbacks), yet relatively little is known about the detailed interplay between ocean warming, ice shelf melting and atmospheric changes. Based on our results, we can also speculate that the projected weakening of the meridional gradient off the Antarctic coasts (Supplementary Figs.7i,l) may have a role since, despite being moderate in the austral summer, it may cause TN variability to decrease (Fig.3d) and thus accelerate TNn warming compared to the seasonal background (Fig.3b). Any such conjecture however should be taken with

caution, primarily because DJF changes here are rather small and model errors large.

→ **L531**: *I'm not sure what is meant by "its fast increase year-round". Are the authors referring to annual mean warming?*

R2.11 The increase here regards variability in all seasons, hence *its fast increase year-round* has been replaced with *its increase in all seasons* (Fig.3c) (**line 621**).

→ **L562**: *Suggest "human-induced" or "anthropogenic" instead of "man-made".*

R2.12 We have replaced *man-made* with *anthropogenic* (**line 658**).

→ **L646**: *Interested in understanding what the justification is for using a 21-year running average to determine the GSAT exceedances? Is this standard practice? Also, is the exceedance year the middle year in question? For example, for ACCESS-CM2 passing 2K in 2038, is this the 2028-2048 average?*

R2.13 While running averages are common tools to filter out noise in time series, our choice of a double decade around each year (21 years in total) is a rather subjective one, though widely used. Its aim is to obtain a smooth enough curve of GSAT changes (i.e., of the average changes over the centered time window) to approximately determine when a specified GWL is passed. The exceedance year is when the smoothed GSAT curve intersects the given GWL and is the middle year of the 21-year period. Any other choice of the time window or smoothing approach, however, likely gives very similar results.

→ **Table S4**: *Is the assumption that all these correlations are statistically significant?*

R2.14 All correlations are highly significant, which is ensured by the large number of data points involved and is now explicitly stated in the **caption of Supplementary Table 4**.

References: The authors have provided a suitable list of prior research.

Reviewer #3

First review of 'Quantifying the role of variability in future intensification of heat extremes', submitted to Nature Communications

** General comments **

*This paper uses the current generation of climate projections (CMIP6) to investigate how both the historical temperature variability and its projected change contribute to the changes in temperature extremes. It is argued that (i) in regions where the historical temperature variability is small (typically the tropics), the future warming will result in more frequent exceedances of present-day high percentiles than elsewhere and that (ii) in regions/seasons where the temperature variability increases, the hot (cold) extremes will warm faster (slower) than the mean, and vice-versa. Such statistical behaviors are qualitatively well known, but here the authors provide several ways to *quantify* them, which seems original to me, and is definitely of high interest to the scientific community.*

*The paper is well written, structured and illustrated, and the supplementary information is helpful to readers interested in more details. Figures are of good quality and contain a lot of information (many panels) — **I would however suggest to enlarge the maps in Figs 1, 2 and 3** which are currently hard to read on a printed version.*

R3.1 As noted in point **R2.2**, we have enlarged Figs.1-3 as much as possible and also changed the color palette in Figs.2,3 to make the maps easier to read.

Overall I really enjoyed reading this paper and I am convinced that it will be of high interest for readers of Nature Comms. However I missed some important methodological details so that I could not fully understand several parts of the paper. In particular **in the text associated with Fig 3 and Table 1, I did not fully understand how TXsd is calculated**, so that I could not follow the discussion and be convinced by the conclusions.

Therefore I am afraid that I can not recommend publication at this stage, because in my opinion **the methodology should be better explained** so that the readers can understand, and in fact reproduce, the analysis made by the authors. Below are some more specific comments and suggestions.

R3.2 We have provided a more in-depth description of the calculation of distribution moments and their changes, including standard deviation, in the *Methods* section.

Specifically, in **lines 767-768** and **780-788** we have clarified the selection of data samples, which consist of annual and seasonal subsets of daily anomalies for every single year of the simulation period, and for every model and grid point. This choice is aimed at tracking the trajectory of distribution moments over time (and with global warming). Furthermore, the grid point series of higher moments are intrinsically detrended since the shifting mean is subtracted year by year by construction, thus avoiding artifacts such as variance inflation (see also point **R3.4** below). To assess changes in distribution moments at specified GWLs relative to the early industrial, we relied on the time averages over the model-specific 21-year period associated with the given GWL (**lines 816-820** in *Methods*).

In order to simplify things further, we have recast **Equation (1)** -relating extremes to distribution moments- more conveniently by replacing derivatives with discrete changes relative to the early industrial era, which changes are displayed in Fig.3. We have rephrased **lines 173-184** describing Equation (1) accordingly, including explicit reference to the *Methods* section for calculation details.

Finally, in **lines 832-838** of *Methods* we have explained more thoroughly the calculation of the coefficients of determination R^2 's (Table 1), by providing concrete examples of how Equations (4) are used together with Equations (1) and (2) to obtain results reported in Table 1 and Fig.7 respectively (see also **lines 226-228** of the main text and point **R3.5** below).

* *Specific comments* *

- **Fig 2 and associated text.** The abbreviations TXx or TNn are traditionally used for annual max/min of daily temperatures in the climate community. I would suggest that the authors better explicit that they use the same abbreviations for seasonal max/min to avoid any confusion. Also, how does the annual cycle affect seasonal TXx or TNn? For instance, in the boreal spring, I would expect the maximum TX in MAM to often occur in late May due to the annual cycle, and I wonder how this kind of features can affect the analysis.

R3.3 We have clarified the use of TXx and TNn for denoting seasonal extremes in the **caption of Fig.2**. With regards to seasonality, we actually removed historical daily normals (the 360/365 day-of-the-year early industrial averages) from model data before carrying out the analyses, as is now better specified in **lines 756,757** of *Methods*. This means that we track the evolution of grid-point daily anomalies irrespective of the local annual cycle. Clearly, seasonality, although removed from the mean of historical distributions (centered about zero), still features their detailed shape, i.e., their variance and skewness.

- **Eq 1 and throughout the text.** It should be further explained how TXsd is computed, i.e. which exact sample is used to compute the standard deviation. Is it the sample of the N years (1851-1900 or the '+2K period') x 90 days (a season) of daily temperature anomalies? Is it detrended so that the long-term warming does not artificially increase the variance?

R3.4 As explained in point **R3.2**, the *Methods* section (**lines 767-768** and **780-788**) now includes a detailed description of the data samples used for calculation of distribution moments and of the rationale for their choice, including the need of removing long-term trends.

In particular, standard deviation is calculated for each annual and seasonal subset of daily anomalies (360 and 90 data at least, respectively) of each year of the simulation period. As moments are central (i.e., about the mean), warming is subtracted year by year. Long-term changes in moments (like in the extremes) are then obtained by the averages over the '+2K period' (or higher), referenced to the early industrial (**lines 816-820**). In such a way, artificial inflation of standard deviation is avoided. Also, as noted again in point **R3.2** above, we have reformulated **Equation (1)** (into integral form) for better clarity.

- **Table 1. It is unclear to me how Eq 4 translates into the values reported here, and how these values are sensitive to the horizontal resolution** — the spatial variance would certainly change for another resolution, but would it affect the relative fractions?

R3.5 As described above (point **R3.2**), we have made more explicit how Equations (4) are used to achieve the coefficients of determination R^2 's in Table 1 by giving specific examples (**lines 832-838** in *Methods* and **lines 226-228** in the main text).

To ascertain whether the R^2 's depend on the horizontal resolution, we repeated the analyses with alternative grids, namely, one coarser (Gaussian N16) than the one we used as a rule (N32) and some finer ones (N64 and N128). That is, for every model we remapped native grid-point changes in distribution moments and extremes to each of the above common grids, before averaging across models. Then we obtained the R^2 's from Equations (4) using multimodel quantities as given by Equation (1), like we did in the N32 case but with varying resolution. Results indicate a little sensitivity of the R^2 's to the horizontal resolution, as is shown in the figure below for TXx warming rates (R^2_{full} only, left). Resolution has also a limited impact on the spatial variance. For example, the JJA distributions of TXx rates (shown below, right) appear very similar in the two edge cases of the N16 and N128 grids, having nearly the same mean and variance. On the other hand, patterns of change (and thus the spatial variance) largely differ across models, irrespective of resolution.

- **Fig 3. There are many acronyms here (and in the first part of the paper) and maybe the most complicated ones such as 'Delta_T TX x m' or 'Delta_EI, T TX x m' could be avoided.**

R3.6 We have simplified notations in Fig.3 by replacing $\Delta_T TXxm$ with the explicit difference $\Delta_T(TXx - TXm)$ and similar for $\Delta_T TNnm$ (panels **a,b,e,f**), so as to not define additional variables. Furthermore, in the legends of panels **e,f** the subscript *EI* is now understood and the use of scaling factors for the excess rates and changes in standard deviation is specified in the figure caption.

- **lines 322-328. Note that all these features are not independent (e.g. the Arctic sea-ice retreat and the reduction in the meridional thermal gradient).**

R3.7 As explained in point **R1.1** above, we have replaced the paragraph in lines 318-337 of the original manuscript with a new section, where we discuss in some detail potential mechanisms behind higher order changes, including winter variability decrease at midlatitudes and its relations with the reduction of the meridional gradient (**lines 343-360**, see also **lines 210-214**).

- **lines 332-337. Note that also zonal thermal gradients are affected under climate change**

(e.g. because land warms faster than ocean), and that it can also contribute to change the temperature variability at a given place. For instance, in Europe, the decrease in winter temperature variability is also associated with a smaller contrast between the climatologically mild ocean and cold continent.

R3.8 In the additional section (see point **R1.1**), we recall the role of zonal gradients with regards to regional temperature variability (**lines 362-375**), and illustrate their changes at the +2K GWL during both winter and summer in **Supplementary Fig.7 (e,f and m,n respectively)**.

- **lines 362-365**. *Surprising because there is a large body of literature (e.g. articles 40, 64 or 65 listed in the references) that suggests an increase in European summer temperature variability due to land-atmosphere feedbacks. Has this increase disappeared in CMIP6?*

R3.9 The paragraph in lines 359-365 could actually be misleading and has been more appropriately rephrased in the revised manuscript (**lines 430-439**). Here we have made clearer that the increase in summer TX_{sd} in the Euro-Mediterranean zone, while still rather small at the +2K GWL, is projected to get larger at higher GWLs as can be seen in Fig.5e (at +3K). This is in fact consistent with previous CMIP5 results (e.g. refs.21,36,39 of the revised manuscript), which are mainly based on simulations of the late 21st century climate when the GWL is typically well above 2K. Furthermore, we recall the most likely mechanisms underneath in **lines 377-381, 385-392** of the new section and **lines 453-457**.

- **line 451**. *Does the 't' actually mean something in the definition of x_+ (e.g. a number of sd) or is it just a typo?*

R3.10 't' is the threshold x_+ expressed in sigma units, $t = x_+/TX_{sd}$ (i.e., the number of sigma x_+ is away from the mean of the unshifted distribution), as is now explicitly stated in **lines 542-543** of the revised manuscript. Also, the paragraph describing Equation (2) has been slightly modified for better clarity (**lines 537-543**).

- **lines 602-605**. *It seems to me that bias correction techniques could easily deal with the issue discussed here, in particular methods that rescale the full pdf/cdf of the historical period. The authors might consider discussing the use of such techniques here.*

R3.11 In the *Discussion* (**lines 688-697**, revised manuscript) we now recall common strategies for constraining model projections of extreme events by observations, including statistical approaches like quantile-based transformations of simulated distributions, and provide in this context a reference (ref.69) to a comprehensive review on the subject of model bias correction.

In addition to the above, we had to make several changes in order to blend seamlessly all the new material into the original manuscript, while keeping the main text within the journal limits. The major ones regard the beginning of the *Hotspot* section (**lines 414-429**), the end three paragraphs of the (original) *Probability* section (shortened to **lines 622-633**), as well as the *Discussion* (e.g., lines **674-676, 682-683, 698-708, 719-723**). We have also made minor adjustments to comply with the journal style (subheading structure of *Methods*) and conventions (figure titles and labels, equations and citation of Supplementary items in the main text).

REVIEWERS' COMMENTS

Reviewer #1 (Remarks to the Author):

Comments addressed satisfactorily.

Reviewer #2 (Remarks to the Author):

Review of "Quantifying the role of variability in future intensification of heat extremes" resubmitted for publication in Nature Comms.

The authors have satisfactorily responded to my questions and suggested edits. In my opinion, the revised manuscript is worthy of acceptance and publication in Nature Comms. I just have a few very minor points below. I commend the authors for this interesting and thorough quantification of future heat extremes. I also like how they have considered the intermodel and intramodel spreads in the Supplementary.

Minor corrections:

L360: Suggest this should be "thereby corroboration".

L408: Thinking here that "targeted" sounds better than "tailor-made".

L435: Is "modest" meant to mean "non-significant" here?

L500: Do the authors mean "one additional degree of global warming" in the same vein as "for every additional degree of warming, we expect..."?

L759: Its unclear to me here what the authors are referring to as "running averages", given they are discussing the daily anomalies.

Answer to the Reviewers

Manuscript NCOMMS-22-13201A, *Quantifying the role of variability in future intensification of heat extremes*

We thank the reviewers for carefully reading our revised manuscript and for their positive comments. We have modified the manuscript to address the remaining concerns of the reviewers. Below we answer on a point-by-point basis to their comments and describe in detail the changes we have made (shown with color highlighting in the manuscript).

Reviewer #2

Review of "Quantifying the role of variability in future intensification of heat extremes" re-submitted for publication in Nature Comms.

The authors have satisfactorily responded to my questions and suggested edits. In my opinion, the revised manuscript is worthy of acceptance and publication in Nature Comms. I just have a few very minor points below. I commend the authors for this interesting and thorough quantification of future heat extremes. I also like how they have considered the intermodel and intramodel spreads in the Supplementary.

Minor corrections:

L360: Suggest this should be "thereby corroboration".

The change has been made.

L408: Thinking here that "targeted" sounds better than "tailor-made".

The change has been made.

L435: Is "modest" meant to mean "non-significant" here?

The term 'modest' relates to the magnitude of the change (at +2K of global warming), not significance, as is now explicitly stated (line 438).

L500: Do the authors mean "one additional degree of global warming" in the same vein as "for every additional degree of warming, we expect...?"

In this context "one additional degree of global warming" simply means +3K of global warming, as is now specified (line 503).

L759: Its unclear to me here what the authors are referring to as "running averages", given they are discussing the daily anomalies.

We used a moving window centered around each calendar day to compute the (smoothed) absolute mean temperature (climate normal) for that day, over 1851-1900. Daily anomalies were then obtained as deviations of the absolute temperatures (model output) from the respective day-of-the-year normals. We have clarified this point in lines 762-763.